# COUP-TFII-mediated reprogramming of the vascular endothelium counteracts tumor immune evasion

Yu Zhu [1,2,4] ✉, Kevin F. Brulois[1,2], Thanh T. Dinh[1,2,3], Junliang Pan [1,2] & Eugene C. Butcher [1,2,4] ✉

T cell scarcity in tumor tissues poses a critical challenge to cancer immunotherapy. Here we manipulate the tumor vasculature, an essential regulator of immune cell trafficking, to reinvigorate anti-tumor T cell responses in "cold" tumors. We show that ectopic pan-endothelial expression of COUP-TFII, a master transcription factor for venous development, induces molecular programs of post-capillary venules in tumor endothelium. Venular reprogramming selectively promotes T cell recruitment into tumors, inhibits tumor growth in mouse models of breast and pancreatic cancers, and sensitizes tumors to immune checkpoint blockade and adoptive T cell transfer therapies. Mechanistic studies show that enhanced recruitment of anti-tumor T cells and tumor inhibition are mediated by COUP-TFII-induced vascular adhesion receptors. Our study supports a pivotal role of vascular endothelial cells in governing tumor immune evasion, and proposes venular reprogramming as a therapeutic strategy to bolster anti-tumor immunity and immunotherapy.

The success of immunotherapies in subsets of cancer has demonstrated the promising capacity of T cells in restraining cancer growth[1,2]. However, immunotherapies are still ineffective in most cancers, especially in patients with immunologically cold tumors, such as breast cancer and pancreatic ductal adenocarcinoma (PDAC)[3,4]. A critical challenge is posed by the inability of antitumor T cells to infiltrate into the tumor tissues even after therapeutic intervention, rendering tumor cells evasive to the immune system[5,6].

Lymphocyte recruitment is tightly regulated in a multistep process that is initiated at post-capillary venules, where the blood vascular endothelial cells (BECs) express adhesion receptors and chemoattractants necessary for the arrest and diapedesis of circulating immune cells[7,8]. In contrast to venular endothelium in other inflammatory conditions, tumor endothelium often lacks proper leukocyte-recruitment functions[9–11], which can promote tumor immune evasion.

Because of its interface with the circulation and its critical roles in tumor growth, the tumor vasculature has long been targeted for tumor therapy. Initial efforts concentrated on impeding the supply of oxygen and nutrients to tumors by targeting pathways essential for endothelial survival and angiogenesis, such as vascular endothelial growth factors (VEGF), angiopoietins, and Tie2[12–15]. However, the clinical efficacy of such strategies has been limited. An alternative approach aims to 'normalize' tumor blood vessels by employing low doses of anti-angiogenic agents to enhance vessel integrity and perfusion, thereby improving the delivery of chemo- or radiotherapy. While successful in certain tumors[16,17], this strategy has shown limited efficacy in other cancer types, such as breast cancer and PDAC.

The classical antiangiogenic therapies that target supply vessels also deplete post-capillary venules, thereby further deterring the recruitment of tumor-restraining lymphocytes and exacerbating the disease for patients with cold tumors. This concept led us to seek alternative approaches to selectively enhance venular functions for lymphocyte recruitment. To this end, we investigated a strategy to reprogram capillary ECs into venular ECs by ectopically expressing COUP-TFII, a member of the orphan nuclear receptor superfamily. During embryonic development, COUP-TFII is required for and drives

[1]Laboratory of Immunology and Vascular Biology, Department of Pathology, Stanford University School of Medicine, Stanford, CA, USA. [2]Palo Alto Veterans Institute for Research, Veterans Affairs Palo Alto Health Care System, Palo Alto, CA, USA. [3] Department of Biological Sciences, San Jose State University, San Jose, USA. [4]These authors contributed equally: Yu Zhu, Eugene C. Butcher. ✉e-mail: zhuyu88@stanford.edu; ebutcher@stanford.edu

venous specification[18]. COUP-TFII expression is retained in adult veins where it is thought to maintain the venous phenotype by suppressing arterial differentiation and promoting venous gene expression[19]. For example, COUP-TFII activates the transcription of ephrin B4 (*Ephb4*) and E-selectin (*Sele*) that mark the identity and functions of veins or venules[18,20,21]. COUP-TFII can also seed transcription factor complexes that control tissue-specific venous expression of vascular addressins MAdCAM1 and St6gal1[22].

Here, we show that ectopic expression of COUP-TFII in endothelial cells enhances the molecular programs and functions of post-capillary venules in tumor EC. COUP-TFII-driven capillary-to-venule reprogramming induces tumor EC expression of lymphocyte adhesion receptors, and promotes T cell recruitment to restrain cancer growth in mouse breast tumor and PDAC. Furthermore, in these treatment-resistant models, EC reprogramming significantly enhances the efficacy of immune checkpoint blockade and adoptive T cell transfer therapies. These data show that reprogramming the endothelium for venular functions not only activates the recruitment of antitumor T cells but also sensitizes cold tumors to immunotherapies.

## Results

### Tumor expansion downregulates T cell infiltration and suppresses venular endothelial specialization

In two mouse models of cold tumors, we characterized the endothelial cell composition and the BEC expression of venular phenotypes associated with leukocyte recruitment. We first used a genetically engineered mouse model (GEMM) that develops spontaneous mammary tumors driven by the mouse mammary tumor virus-polyoma middle T antigen (MMTV-PyMT)[23]. We dissociated tumor tissues from 4-month-old MMTV-PyMT mice and performed flow cytometry to analyze the abundance of BECs in tumors smaller than 0.2 g of wet weight, and compared with tumors larger than 0.5 g of weight. The frequency of BECs was not affected by tumor size: in both small and large tumors, BECs accounted for ~2% of cells dissociated from the tumor (Fig. 1A, B). However, the percent of BECs expressing the leukocyte adhesion molecule P-selectin[24-27] decreased significantly in larger tumors (Fig. 1C, D). The mean expression of venule markers ICAM1[28,29] and CD157[30] in BECs also significantly decreased (Fig. 1E, F), suggesting an overall reduction in features associated with post-capillary venules. Similarly, in an orthotopic PDAC mouse model established using a syngeneic PDAC cell line from p48-Cre, Kras$^{G12D}$, p53$^{m/m}$ mice (termed KPC hereafter)[31], we also observed decreased frequencies of P-selectin+ BECs and reduced expression of leukocyte adhesion molecules as tumors expanded in size (Supplementary Fig. 1A).

The decreased expression of venular BEC molecules in larger tumors correlated with significantly decreased frequencies of tumor-infiltrating T cells, including both CD8+ and CD4 + T cell subsets, and a decreased effector-to-Treg ratio (Fig. 1G–J and Supplementary Fig. 2A). On the other hand, expansion in tumor size did not correlate with changes in the frequencies of tumor-associated macrophages (TAMs) or neutrophils (granulocytic myeloid derived suppressor cells, or G-MDSCs), the two most abundant immune populations infiltrating the tumor tissue (Fig. 1K–N and Supplementary Fig. 2B). Thus, the loss of vascular selectin-expressing venules during tumor expansion selectively correlates with decreased T cell infiltration into the tumor tissue.

Given the importance of venules in the regulation of leukocyte recruitment, we hypothesized that strategies to expand venular ECs could reverse the decrease in T cell infiltration, and could be beneficial for treating immunologically cold tumors that are scarce in T cells.

### Ectopic expression of COUP-TFII in ECs drives capillary-to-venule reprogramming

The transcription factor COUPT-TFII (encoded by the *Nr2f2* gene) is a master regulator of venous differentiation during embryonic development[18]. scRNAseq of tumor BECs confirmed selective expression of *Nr2f2* by venular BECs in both PyMT and KPC tumors (Supplementary Fig. 1B–D, G–I). *Nr2f2* also correlates with expression of venule-defining molecules, including genes encoding leukocyte adhesion receptors vascular cell adhesion molecule-1 (VCAM1), ICAM1, P-selectin, E-selectin, ephrin type-B receptor 4, and CD157 (Supplementary Fig. 1E). On the other hand, *Nr2f2* expression inversely correlates with the expression of genes that mark capillaries or arteries, such as *Notch4*, *Hey1*, *Nrp1*, *Efnb2*, *Icam2*, and *Ly6c1* (Supplementary Fig. 1F). Therefore, we hypothesized that ectopic expression of COUP-TFII in capillary tumor endothelial cells (TECs) could promote venous differentiation to enhance immune cell recruitment into the tumor tissue.

To ectopically express COUP-TFII in tumor capillary BECs, we established a mouse model that drives inducible expression of COUP-TFII specifically in ECs, by crossing the VECadherin-Cre$^{ERT2}$ strain[32] with the Rosa26$^{LoxP-Stop-LoxP(LSL)-COUP-TFII}$ mice[33] (termed iCoup mice hereafter). In these mice, we established orthotopic mammary tumors with a syngeneic cancer cell line derived from autochthonous tumors of MMTV-PyMT mice (both on the C57BL/6 background). Approximately 7 days after tumor implantation, when tumors became palpable, we treated mice with tamoxifen to induce ectopic COUP-TFII expression in capillary BECs. Two weeks later, we dissociated the tumors and performed flow cytometry to quantify the expression of capillary and venular markers in BECs. By flow cytometry, iCoup led to a sixfold increase in the mean fluorescence intensity of intracellular COUP-TFII in the blood endothelial cells (Fig. 2A, B), but not in other cells. Immunofluorescence imaging also confirmed widespread induction of COUP-TFII expression selectively in the tumor endothelium but not in other cell types (Fig. 2C). While COUP-TFII-expressing venules were occasionally observed in control tumors, iCoup led to extensive COUP-TFII expression in tumor endothelium, in both intratumor and peritumor regions (Supplementary Fig. 3A). Upon ectopic COUP-TFII expression, a larger fraction of BECs adopted the phenotype of P-selectin+ venules (Fig. 2D, E). We also saw increased expression of venular markers ICAM1, MAdCAM1 (mucosal vascular addressin cell adhesion molecule-1), CD157, and VCAM1 (Fig. 2F–I, M, N). On the other hand, iCoup significantly downregulated BEC expression of capillary markers, such as podocalyxin (Podxl) and Ly6C (Fig. 2J, K). The expression of EC identity marker CD31 remained unaffected (Fig. 2L, M). We observed similar shifts in capillary and venular marker expression patterns in orthotopic and subcutaneous KPC tumors (Supplementary Fig. 4A). These data suggest that ectopic expression of COUP-TFII in ECs led to a loss of the capillary signature coupled with upregulated expression of venular identity markers and leukocyte adhesion molecules. The de novo programmed venular ECs, marked by MAdCAM1 that is not expressed in control mice but induced by iCoup, can be observed in the tumor nests, peritumoral regions, and the stromal area distant from malignant cell-enriched regions (Supplementary Fig. 3B). In the adjacent normal pancreas, we observed endothelial cells with prominent features of arterial blood vessels despite ectopic expression of COUP-TFII (Supplementary Fig. 3C), suggesting that ectopic COUP-TFII expression in adult mice may not be sufficient to reprogram the established arterial ECs into venules or veins.

Notably, iCoup significantly enhanced the staining of BECs by the F2 antibody[34] (Fig. 2O, P), which recognizes Sialyl Lewis X decorated with α2-3-sialylation and α1-3-fucosylation. The F2 epitope is a glycotope precursor of peripheral node addressin (PNAd). However, iCoup did not induce the expression of mature PNAd in the BECs of PyMT and KPC tumors, as indicated by a lack of MECA79 antibody staining (Fig. 2Q and Supplementary Fig. 4A). These data suggest that Coup-TFII induction was not sufficient to confer a PNAd+ high endothelial venule (HEV) identity, but was sufficient to enhance endothelial programs associated with post-capillary venules at the expense of capillary ECs.

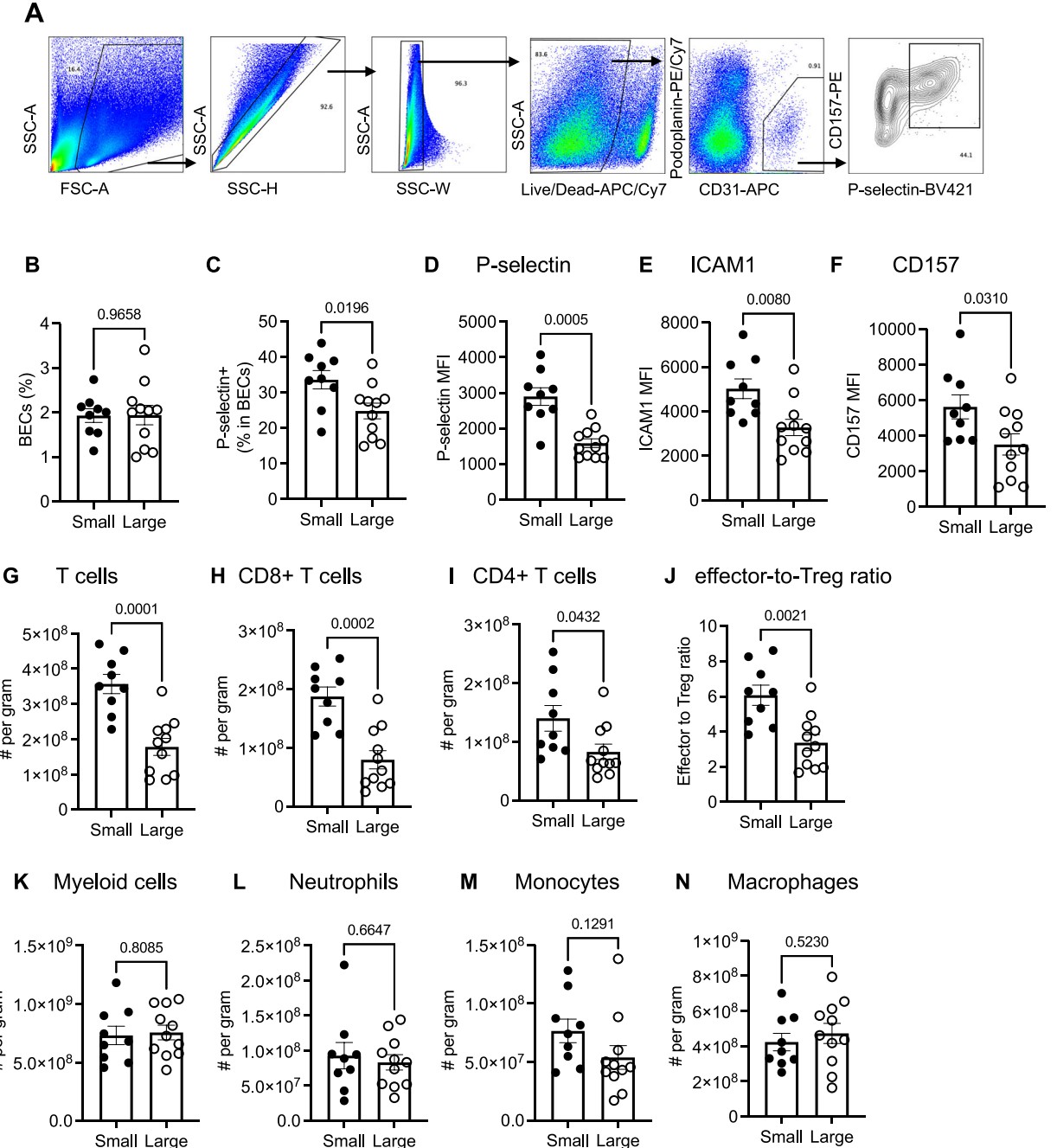

**Fig. 1 | Tumor expansion downregulates T cell infiltration and venular endothelial specialization. A** Representative flow cytometry gating strategy for vascular blood endothelial cells (BEC) in MMTV-PyMT tumors. **B**, **C** Flow cytometric quantification of tumor-associated BEC abundance. **D**–**F** Quantification of mean fluorescence intensity (MFI) of indicated markers in tumor BECs. **G**–**N** Flow cytometric quantification of the frequencies of indicated leukocytes in autochthonous MMTV-PyMT-B6 tumors from 4-month-old mice. **B**–**N** small: *n* = 9 biological replicates; large: *n* = 11 biological replicates). Representative of two independent experiments. Groups were compared by an unpaired two-tailed Student's *t*-test. Error bars indicate s.e.m.

We also assessed the impact of ectopic COUP-TFII on BECs in other tissues 2 weeks after tamoxifen induction. In the lung, ectopic COUP-TFII led to decreased fluorescence intensity of P-selectin on BECs, while the expression of ICAM1, MAdCAM1, and Ly6C remained unaltered (Supplementary Fig. 4B). In the liver, iCOUP upregulated the expression of ICAM1 but not MAdCAM1 or P-selectin (Supplementary Fig. 4C). BECs in the kidney did not change their cell surface expression of these venular markers, but Ly6C was slightly downregulated (Supplementary Fig. 4D). These data, along with our previous studies of blood endothelium in lymphoid organs, suggest that the impact of ectopic COUP-TFII expression on endothelial phenotypes is dependent on the tissue context. As the tumor EC are activated and angiogenic, they may be particularly susceptible to reprogramming.

To determine whether this capillary-to-venule reprogramming in tumors occurred in a cell-autonomous manner, we induced ectopic COUP-TFII expression in a subset of ECs to compare the phenotypes of iCoup cells with uninduced control BECs. Towards that end, we generated heterozygous Rosa26$^{LSL-iCOUP}$ x Rosa26$^{LSL-tdTomato}$ inducible reporter mice driven by Cdh5-Cre$^{ERT2}$, implanted PyMT or KPC tumors

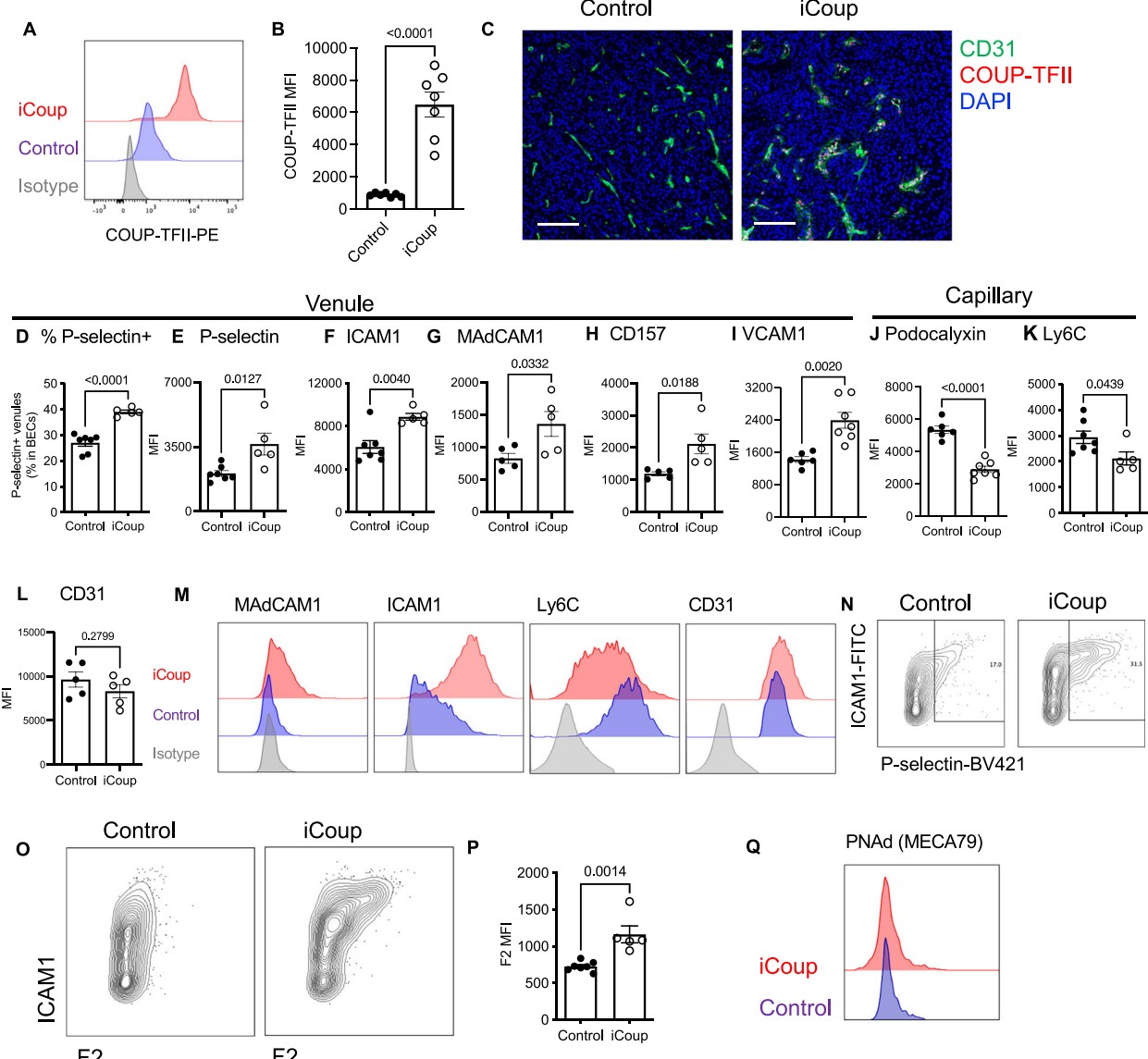

**Fig. 2 | Ectopic expression of COUP-TFII in endothelial cells drives capillary-to-venule reprogramming. A, B** Representative histogram and MFI quantification of COUP-TFII in BECs from orthotopic PyMT tumors established in iCoup and control mice. Pre-gated on BECs. Representative of three independent experiments. ($n = 7$ biological replicates/group). **C** Representative immunofluorescence imaging of COUP-TFII and CD31 in orthotopic PyMT tumors established in iCoup and control mice. Scale bar represents 50 μm. **D** Flow cytometry quantification of frequency of P-selectin+ venular BECs. **E–L** MFI of indicated markers in BECs from orthotopic PyMT tumors established in iCoup and control mice (**D–F, I–K** control: $n = 7$ biological replicates, iCoup: $n = 5$ biological replicates; **G, H, L** $n = 5$ biological replicates/group). **M–O** Representative histograms and contour plots of indicated molecules in BECs from orthotopic PyMT tumors of iCoup and control mice. Pre-gated on BECs. **P** MFI of F2 antibody staining in BECs from orthotopic PyMT tumors established in iCoup and control mice (Control: $n = 7$ biological replicates, iCoup: $n = 5$ biological replicates). **Q** Representative histograms of PNAd in BECs from orthotopic PyMT tumors of iCoup and control mice. Pre-gated on BECs. Data in (**D–N**) are representative of more than five independent experiments. Data in (**O–Q**) are representative of three independent experiments. Groups were compared by unpaired two-tailed Student's *t*-test. Error bars indicate s.e.m.

orthotopically, and then treated tumor-bearing mice with limiting doses of tamoxifen (Fig. 3A). Because LSL-tdTomato and LSL-iCoup are both in the Rosa26 locus and LoxP cleavage is primarily limited by CreERT2 induction levels, we anticipated that tdTomato and iCOUP would be co-induced. Our dosing strategy (7.5 mg per kilogram of body weight) labeled on average 50% of the BECs with tdTomato (Fig. 3B), allowing us to compare BECs enriched for ectopic COUP-TFII (tdTomato+) with control BECs (tdTomato-) internally in the same mouse. Pairwise comparisons by flow cytometry showed that venular markers (ICAM1, CD157, P-selectin) were significantly higher in tdTomato(+) BECs than in the tdTomato(−) counterpart, while capillary marker Ly6C was lower in tdTomato(+) cells (Fig. 3C). As a control, the

expression of the pan-endothelial cell identity marker CD31 was similar in tdTomato(+) and (−) BECs (Fig. 3C).

In addition, we extracted mRNA from sorted tdTomato(−) and (+) BECs, and performed a quantitative polymerase chain reaction (QPCR) array. By pairwise comparisons, tdTomato(+) cells had higher expression of *Nr2f2* than cells lacking the reporter expression, validating the approach of using tdTomato as a surrogate marker for ectopic Nr2f2 expression. tdTomato-expressing BECs showed elevated expression of molecules encoding venular genes *Ephb4*, *Sele*, and *Madcam1* (Fig. 3E), but decreased expression of genes associated with capillary identity and functions, including *Rbpj* and *Podxl* (Fig. 3F). As a control, vascular endothelial identity markers *Cdh5* and *Eng* (endoglin) were not

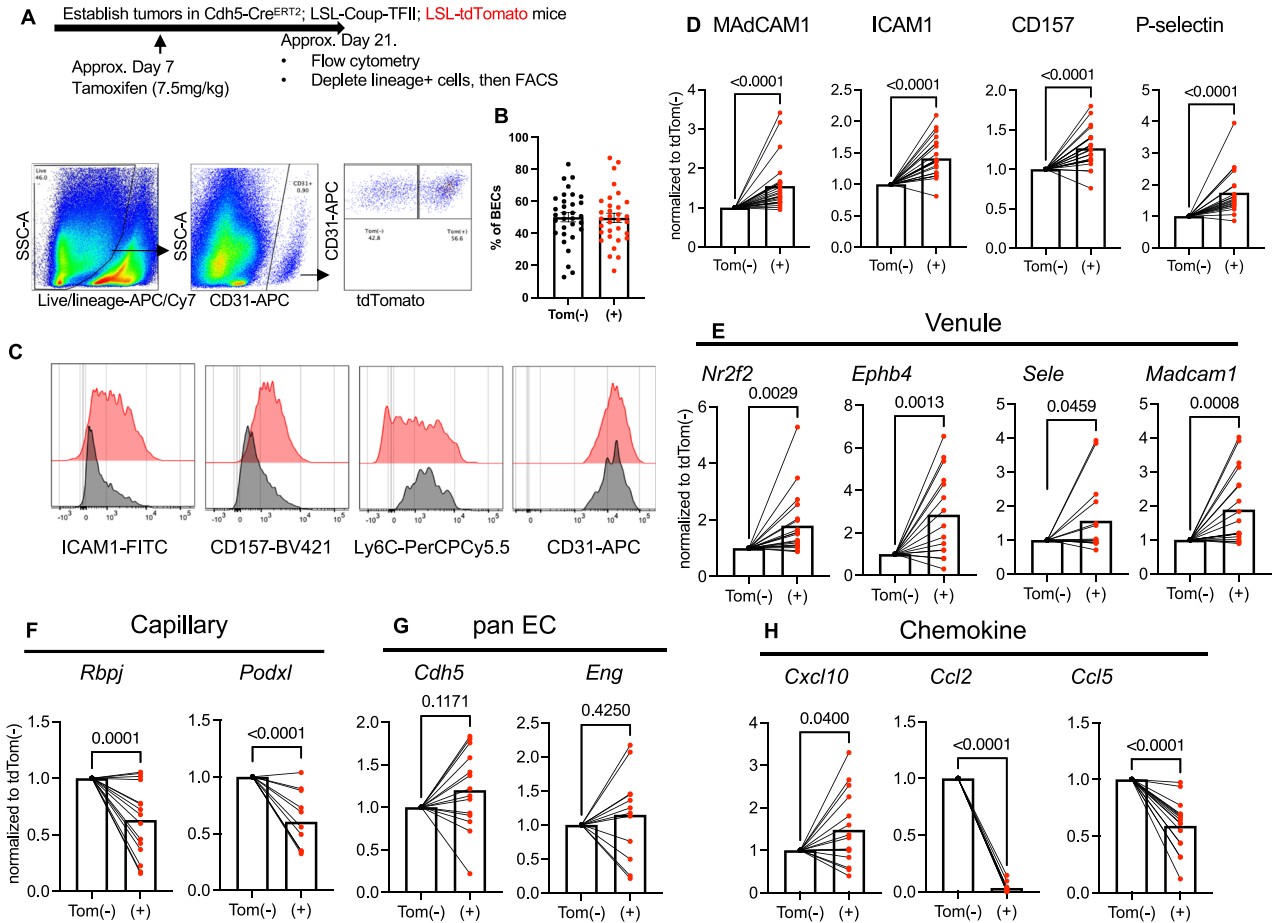

**Fig. 3 | Ectopic expression of COUP-TFII reprograms ECs in a cell-autonomous manner. A** Schematics of mosaic COUP-TFII induction in tumor BECs and gating strategy. Lineage+ (Ter119+/EpCAM+/CD45+/Podoplanin+) cells were depleted from digested tumors before proceeding to fluorescence-activated cell sorting (FACS). **B** Quantification of tdTomato (Tom+) labeling efficiency in (**A**). (*n* = 32/group). **C**, **D** Representative histograms (**C**) and MFI quantification (**D**) of indicated marker expression in tdTomato(−) and (+) BECs from orthotopic tumors; MFI values of tdTom(+) BECs were normalized internally to those of the tdTom(−) counterpart within the same mouse, and pairwise comparison was performed (*n* = 25 biological replicates/group for MAdCAM1 and CD157, *n* = 23 biological

replicates/group for ICAM1, *n* = 13 biological replicates/group for P-selectin). Representative of three independent repeats. **E**–**H** QPCR analyses of indicated molecules of tdTomato(−) and (+) BECs sorted from orthotopic tumors; gene expression values from each mouse were normalized internally to that of the tdTomato(−) BECs and pairwise comparison was performed (*n* = 19 biological replicates/group for *Nr2f2* and *Madcam1*; *n* = 16 biological replicates/group for *Ephb4*, *Sele* and *Ccl5*; *n* = 15 biological replicates/group for *Rbpj* and *Cxcl10*; *n* = 12 biological replicates/group for Eng, *n* = 11 biological replicates/group for *Podxl* and *Ccl2*; *n* = 14 biological replicates/group for Cdh5). Groups were compared by unpaired two-tailed Student's *t*-test. Error bars indicate s.e.m.

different between tdTomato(+) and (−) BECs (Fig. 3G). Intriguingly, tdTomato(+) BECs had elevated expression of *Cxcl10*, encoding a chemokine that promotes the recruitment of CD8+ and effector CD4 + T cells[35–37]. Conversely, tdTomato(+) BECs had lower expression of myeloid-recruiting chemokines *Ccl2* and *Ccl5* than the tdTomato(−) counterpart[38–41] (Fig. 3H). Taken together, the mosaic labeling experiments indicate that ectopic expression of COUP-TFII drives a loss of capillary identity and upregulates molecular programs of post-capillary venules in a cell-autonomous manner.

We did not observe an increased abundance of BECs upon ectopic COUP-TFII expression (Supplementary Fig. 5A). Ectopic COUP-TFII expression in ECs also did not change the coverage of tumor vessels by pericytes, identified as NG2-expressing perivascular cells (Supplementary Fig. 5B, C). To determine whether ectopic Coup-TFII expression changed capillary functions, we evaluated oxygenation levels in the tumor by assessing hypoxia. Consistent with decreased capillary marker expression on BECs, tumors in iCoup mice had significantly elevated levels of hypoxia (Supplementary Fig. 5D, E), suggesting that ectopic COUP-TFII in BECs led to compromised capillary functions.

## COUP-TFII-driven capillary-to-venule reprogramming enhances T cell recruitment

We next investigated the impact of iCoup-mediated EC reprogramming on leukocyte recruitment. Towards this end, we performed flow cytometry to quantify the abundance of tumor-infiltrating myeloid and lymphoid cells. In iCoup mice, we saw a significant increase in the frequency of tumor-infiltrating T cells, including both CD8+ and CD4 + T cells (Fig. 4A–D). On the other hand, tumor-infiltrating myeloid cells did not increase in iCoup mice (Fig. 4E): the abundance of monocytes, eosinophils, and neutrophils was not affected by iCoup (Fig. 4F, G), while the percentage of tumor-associated macrophages (TAMs) decreased slightly (Fig. 4H). The abundance of tumor-infiltrating CD103+ conventional dendritic cells (cDC1) also decreased in iCoup mice, while myeloid DCs (cDC2) did not change (Fig. 4I, J and Supplementary Fig. 2C). The abundance of B cells was also not affected in the tumor tissue (Fig. 4K). Similarly, in autochthonous tumors of the MMTV-PyMT GEMM, induction of ectopic COUP-TFII in BECs also increased T cell abundance (Fig. 4L, M). Additionally, in both orthotopic and subcutaneous KPC tumor models, iCoup also increased the frequencies of tumor-infiltrating T cells but

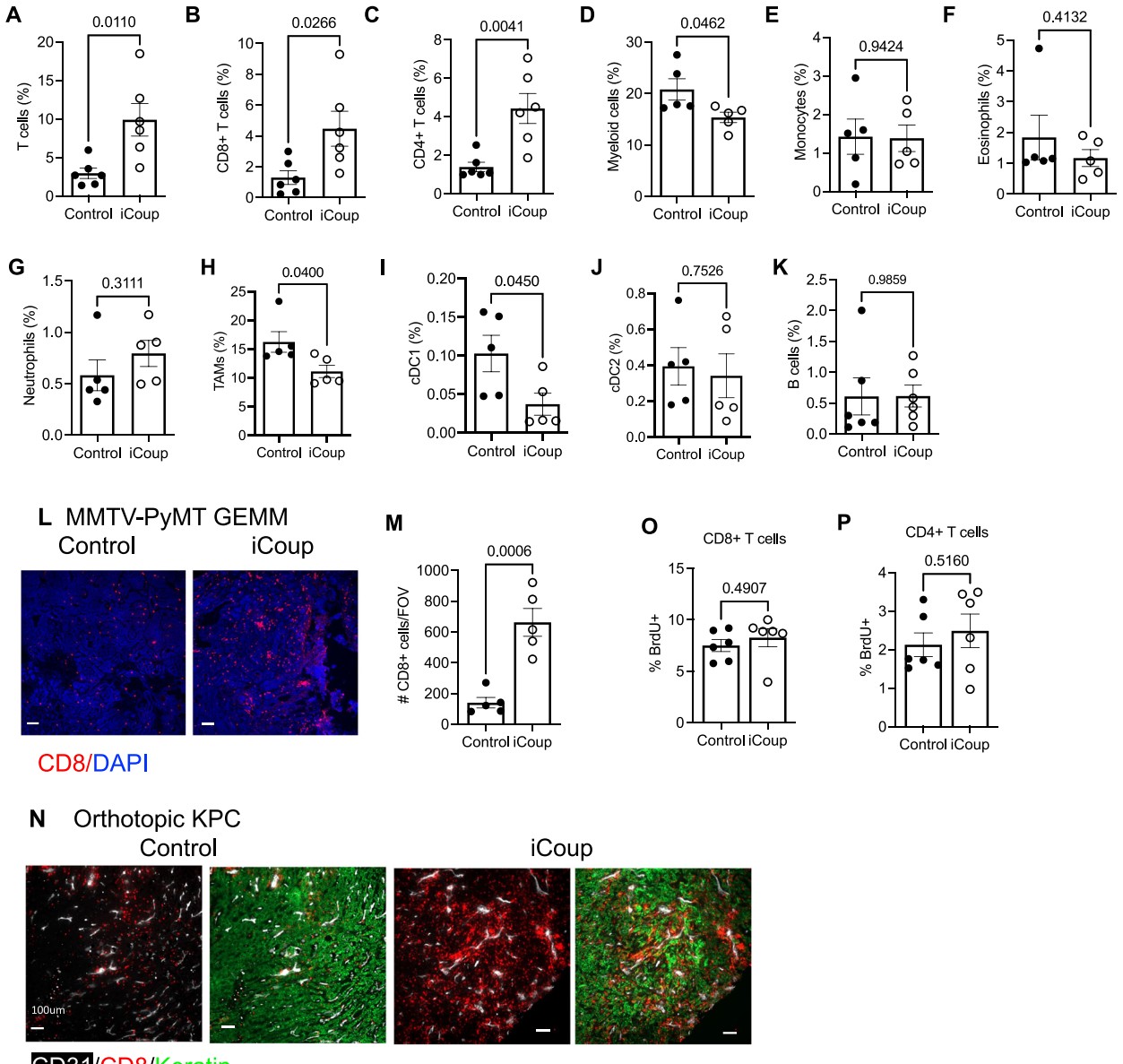

**Fig. 4 | COUP-TFII-driven capillary-to-venule reprogramming enhances T cell infiltration. A–C** Flow cytometric quantification of the frequency of indicated lymphocytes among cells dissociated from orthotopic PyMT-B6 tumors in iCoup and control mice, based on gating strategies in Supplementary Fig. 2; representative of three independent experiments. (*n* = 6 biological replicates/group). **D–K** Flow cytometric quantification of the frequency of indicated leukocytes among cells dissociated from orthotopic PyMT-B6 tumors in iCoup and control mice, based on gating strategies in Supplementary Fig. 2; representative of three independent experiments. (*n* = 5 biological replicates/group). **L, M** Representative immunofluorescence imaging (**L**) and quantification (**M**) of CD8+ cells in GEMM tumors from 5-month-old MMTV-PyMT-iCoup and control MMTV-PyMT mice. Scale bars represent 100 μm. (*n* = 5 biological replicates/group). **N** Representative immunofluorescence imaging of CD8, CD31, and keratin of orthotopic KPC tumors established in control and iCoup mice. Scale bars represent 100 μm. **O, P** Flow cytometric quantification of BrdU incorporation in tumor-infiltrating CD8+ and CD4 + T cells. Representative of two independent experiments. (*n* = 6 biological replicates/group). Indicated cell populations are quantified as % of total cells dissociated from the tumor, based on gating strategies from Supplementary Fig. 2. Groups were compared by unpaired two-tailed Student's *t*-test. Error bars indicate s.e.m.

not the myeloid cell populations (Supplementary Fig. 6A, B). Many T cells are adjacent to endothelial cells and are in contact with tumor cells (Fig. 4N). Taken together, the changes in the tumor immune profile across multiple cancer models suggest that COUP-TFII-driven capillary-to-venule reprogramming selectively promoted the infiltration of T cells into the tumor tissue.

We also evaluated whether ectopic COUP-TFII expression in vascular beds other than the tumors had an impact on the local immune microenvironment. In contrast to our observations in the PyMT and KPC tumors, the abundance of T cells in the lung decreased in iCoup

mice 2 weeks after tamoxifen induction, while the frequency of myeloid cells increased (Supplementary Fig. 4E). We did not observe significant alterations in the frequencies of liver-infiltrating T cells or myeloid cells (Supplementary Fig. 4F). In the kidney, the abundance of T cells also decreased while the myeloid cells were not significantly changed (Supplementary Fig. 4G). These data suggest that the impact of ectopic COUP-TFII expression on the local immune milieu is tissue-specific; the underlying mechanisms require further dissection.

We then asked how iCoup compares with antiangiogenic reagents in reprogramming endothelium and inducing T cell responses.

Towards that end, we treated orthotopic PyMT or KPC tumor-bearing mice with antibodies against VEGF receptor-2 (VEGFR2) at 40 mg/kg or 10 mg/kg, two doses previously shown to deplete or normalize tumor vasculature respectively[42]. In PyMT tumors, both doses of VEGFR2 inhibition decreased the abundance of P-selectin+ venular ECs (Supplementary Fig. 6C). At 10 mg/kg, VEGFR2 inhibition did not change the abundance of T cells, and the higher dose of 40 mg/kg treatment significantly decreased the abundance of CD8 + T cells and slightly decreased CD4 + T cells (Supplementary Fig. 6D). The myeloid populations in the PyMT tumors were not significantly altered upon VEGFR inhibition (Supplementary Fig. 6D). In KPC tumors, VEGFR2 inhibition also did not increase the abundance of P-selectin+ venular BECs or T cell infiltration (Supplementary Fig. 6E, F). However, normalizing doses of anti-VEGFR2 decreased the frequencies of tumor-infiltrating neutrophils and macrophages, suggesting that vessel normalization may inhibit tumor growth by alleviating myeloid-mediated immune suppression (Supplementary Fig. 6F), but not by directly enhancing T cell recruitment. These data suggest that COUP-TFII-induced capillary-to-venule reprogramming could present a paradigm of vessel-targeting approaches different from antiangiogenic therapies.

We asked whether ectopic COUP-TFII-driven EC reprogramming increased T cell abundance in the tumor through enhanced in situ T cell expansion. However, 5-bromo-2-deoxyuridine (BrdU) incorporation assay indicated that tumor-infiltrating T cells were not more proliferative in iCoup mice (Fig. 4O, P).

To determine whether the increase in tumor-infiltrating T cells was due to venule-mediated recruitment, we performed short-term homing assay in three independent experimental systems. First, we implanted orthotopic PyMT tumors in a cohort of donor mice and isolated leukocytes from the tumor-draining lymph nodes. We then labeled these cells with carboxyfluorescein succinimidyl ester (CFSE) and injected into a cohort of recipient PyMT-tumor-bearing iCoup or control mice. Sixteen hours after injection, we dissociated tumors of recipient mice, and quantified the abundance of CFSE+ donor-derived T cells that have homed to the tumor tissue. This experimental system mimics lymphocyte homing from draining lymph nodes into tumor tissues. In this setup, ectopic expression of COUP-TFII in ECs significantly upregulated the frequency of donor-derived T cells that were recruited into recipient tumors (Fig. 5A).

In a second system, we established orthotopic PyMT tumors that express ovalbumin (OVA) as a surrogate tumor antigen, and induced ectopic COUP-TFII expression in ECs with tamoxifen. Three days after induction, we adoptively transferred ex vivo-activated CFSE-labeled OT1 T cells, whose T cell receptors are cognate for the OVA antigen. Sixteen hours later, when we harvested the tissues, we found significantly more CFSE-labeled OT1 cells recruited into the tumor tissue (Fig. 5B). Enhanced OT1 homing was also seen in orthotopic PDAC tumors established using a KPC cell line that expresses OVA (Supplementary Fig. 6G).

In a third setup, we injected PyMT tumor-bearing mice with bone marrow cells from donor Rosa26mTmG mice, whose cells carry tdTomato fluorescence, and quantified the number of myeloid cells and T cells that homed to the tumor sixteen hours after transfer. iCoup did not change the recruitment of tdTomato+ bone marrow-derived monocytes, eosinophils, or neutrophils (Fig. 5C). However, more donor-derived T cells were recruited (Fig. 5C), consistent with the changes in the endogenous immune infiltration profile.

We next assessed whether ectopic COUP-TFII expression enhances the ability of BECs to support T cell migration towards tumor cells in an in vitro system without confounding factors from the complex tumor microenvironment. We used bEND3 cells, a capillary microvascular cell line widely adopted for endothelial cell culture studies[43], and established a stable bEND3 line that overexpresses full-length COUP-TFII (Nr2f2) through lentiviral infection. We plated bEND3-COUP-TFII or control vector-transduced bEND3 cells on the transwell

inserts, and quantified the ability of the endothelial cell monolayers to support T cell transwell migration to PyMT tumor cells in the bottom wells. Consistent with the in vivo homing assays, ectopic COUP-TFII expression significantly upregulated the transendothelial recruitment of T cells (Supplementary Fig. 6H).

iCoup mice had an increased percentage of neutrophils and a decreased percentage of B cells in the blood than the control mice, while the frequencies of Ly6C+ and Ly6C− blood monocytes and of T cells were not significantly affected (Fig. 5D, E). Moreover, the frequencies of CD8+ and CD4 + T cells, and the relative abundance of naïve T cells (CD44-CD52L+), effector/memory T cells (CD44+CD62L−), and central memory T cells (CD44+CD62L+) cells was unaffected in the tumor-draining lymph nodes (Fig. 5F–Q). While we could not rule out the contribution of draining lymph nodes to the immune phenotypes observed in iCoup mice, these data suggest that the enhanced T cell infiltration into the tumor is not due to an overall increase in T cells available for recruitment.

Collectively, these studies suggest that COUP-TFII-driven capillary-to-venule reprogramming selectively enhanced the recruitment of T cells to the tumor.

## COUP-TFII-driven endothelial reprogramming inhibits tumor growth via selectin- and chemokine-mediated vessel-T cell interactions

We then assessed the impact of endothelial reprogramming on tumor burden. In both subcutaneous and orthotopic PyMT models, ectopic COUP-TFII expression in ECs significantly decreased tumor weights (Fig. 6A, B). We also assessed the impact of EC reprogramming on autochthonous mammary tumors in the MMTV-PyMT, Cdh5CreERT2, LSL-COUP-TFII GEMM: we induced ectopic COUP-TFII expression in 3-month-old mice, at an age when mice were beginning to develop tumors but before tumors became palpable. Two months after COUP-TFII induction, when mice reached 5 months of age, we found that ectopic COUP-TFII in ECs led to a 54% reduction in tumor weights (Fig. 6C). We also asked if iCoup affected the duration of mouse survival in this GEMM in a second cohort of mice. The median survival of control mice was 25 weeks, while induction of COUP-TFII in ECs extended the survival of tumor-bearing mice to 38 weeks (Fig. 6D). Similarly, in subcutaneous and orthotopic KPC tumor models, iCoup also significantly decreased tumor burden (Fig. 6G, H).

Next, we assessed whether the increased T cell recruitment was necessary for tumor inhibition by depleting CD8 + T cells using CD8β-depleting antibodies. CD8 + T cell depletion in control mice did not significantly inhibit the growth of PyMT tumors, suggesting that on the baseline level CD8 + T cell functions were largely repressed. However, loss of CD8 + T cells in iCoup mice abolished tumor inhibition in this breast cancer model (Fig. 6E, F). Similarly, in orthotopic KPC tumors, T cell depletion also reversed iCoup-mediated tumor inhibition (Fig. 6I). These data suggest that the enhanced recruitment of T cells was necessary for restraining tumor expansion.

Enhanced T cell infiltration in iCoup mice correlated with increased BEC expression of multiple vascular adhesion receptors implicated in leukocyte recruitment, including E- and P-selectins, ICAM1, and VCAM1 (Figs. 2B, C, F, 3C–E). While ICAM1 and VCAM1 are also expressed outside the vasculature (e.g., by cancer-associated fibroblasts and myeloid cells), the vascular selectins are exclusively expressed by endothelial cells and (for P-selectin) platelets, but not on tumor cells or other stromal populations. Therefore, to specifically block venular functions and assess the impact on leukocyte homing, without directly disrupting other stromal populations, we treated PyMT-tumor-bearing mice with antibodies against E- and P-selectin. Interestingly, selectin blockade abolished the iCoup-induced increase in tumor-infiltrating CD4+ and CD8 + T cells but had no significant effects in the control tumor-bearing mice (Fig. 7A, B). Moreover, blockade of E- and P-selectins reversed the tumor inhibition seen in

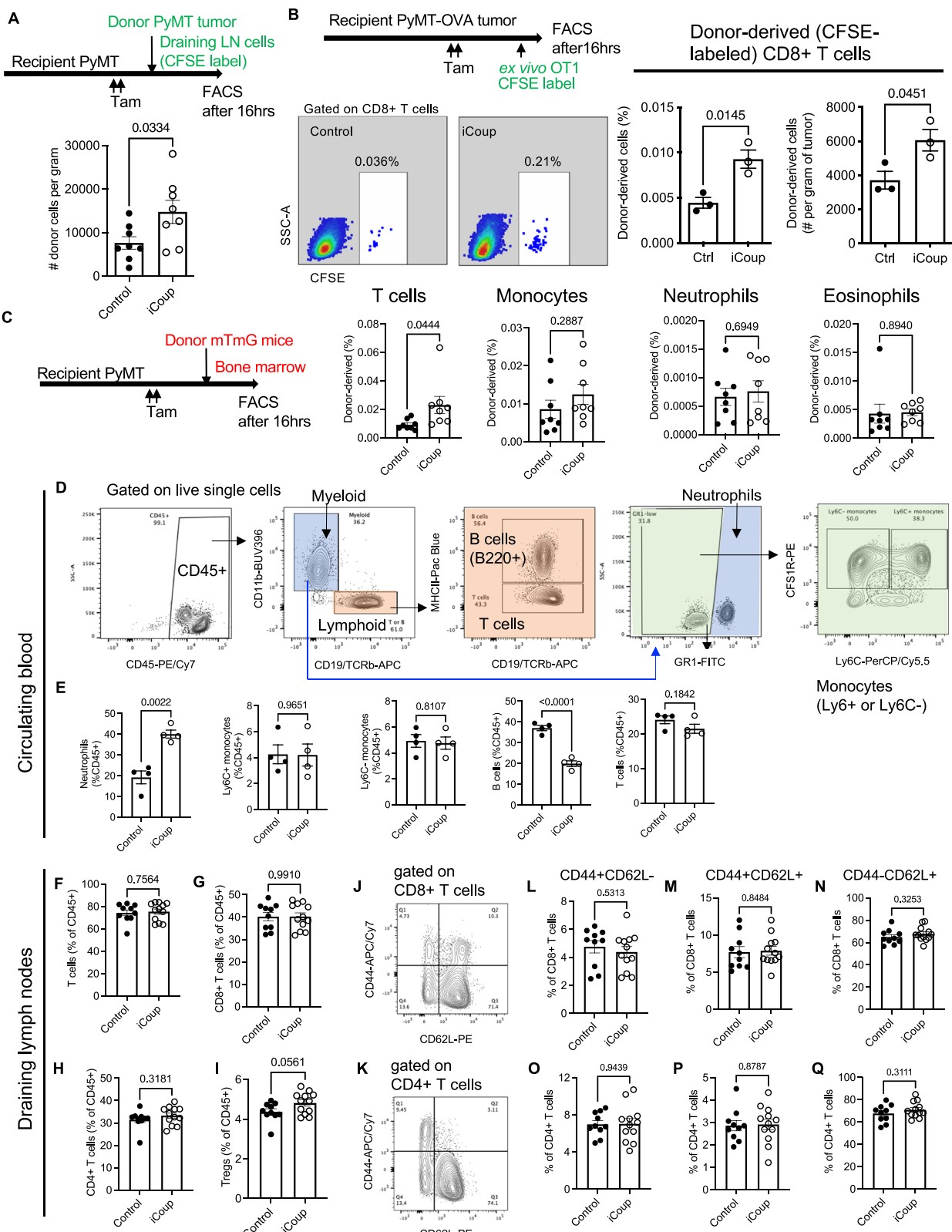

iCoup mice (Fig. 7C), phenocopying the effects of T cell depletion. Given the endothelium-restricted expression of the vascular addressins, these data suggest that ectopic COUP-TFII-mediated EC reprogramming directly promoted T cell recruitment to inhibit tumor burden, and that this effect is mediated by ECs.

Leukocyte recruitment is regulated by the combined actions of vascular adhesion mechanisms and chemoattractant cytokines. Given

the observation that ectopic COUP-TFII upregulated the expression of CXCL10 in tumor ECs in a cell-autonomous manner (Fig. 3H), we hypothesized that this chemokine selectively promoted T cell recruitment: CXCL10 is well established for its role in mediating T cell recruitment through its receptor CXCR3, which is predominantly expressed on activated CD8 + T cells and Th1-differentiated CD4 + T cells[36]. Treatment of PyMT-OVA tumor-bearing mice with

**Fig. 5 | COUP-TFII-driven capillary-to-venule reprogramming enhances T cell recruitment from circulation. A** Schematics and quantification of lymph node T cell homing assay in PyMT-B6 tumors. (*n* = 8 biological replicates/group). Representative of more than three independent experiments. **B** Schematics, representative flow cytometry plots, and quantification of OT1 T cell homing to orthotopic PyMT-OVA tumors. (*n* = 3 biological replicates/group). Representative of three independent experiments. **C** Flow cytometric analyses of bone marrow cell homing to orthotopic PyMT tumors. (*n* = 8 biological replicates/group). Representative of more than three independent experiments. Quantification in (**B**, **C**) indicates the frequency of homed donor-derived leukocytes as % of total cells

dissociated from the tumor. **D** Gating strategy for circulating leukocytes in the blood. **E** Frequencies of circulating leukocyte subsets in the blood of iCoup and control mice, quantified as the percentage among circulating CD45+ cells. (*n* = 4 biological replicates/group). **F–I** Frequencies of indicated T cell populations in the tumor-draining lymph nodes (dLNs). (control: *n* = 10 biological replicates, iCoup: *n* = 12 biological replicates). **J–Q** Representative gating strategy and flow cytometric quantifications of indicated T cell subsets in the dLNs. (control: *n* = 10 biological replicates, iCoup *n* = 12 biological replicates). Groups were compared by an unpaired two-tailed Student's *t*-test, ns denotes not significant. Error bars indicate s.e.m.

CXCR3 blocking antibodies abolished the iCoup-mediated increase in T cell frequency (Fig. 7D, E). On the other hand, CXCR3 blockade did not affect the abundance of TAMs. Importantly, upon CXCR3 blockade, iCoup-mediated tumor inhibition was abolished (Fig. 7F), phenocopying the effect of T cell depletion.

Collectively, these data indicated that ectopic COUP-TFII in tumor endothelium enhances the expression of leukocyte vascular adhesion molecules and chemokines that promote the recruitment of T cells, resulting in inhibition of tumor growth.

## COUP-TFII-driven capillary-to-venule reprogramming sensitizes immune checkpoint blockade and adoptive T cell transfer therapy

Immune checkpoint blockade strategies have shown promising effectiveness in several types of cancer, but their clinical benefits have not translated to most cold tumors. We hypothesized that venular reprogramming of TECs, by enhancing antitumor T cell recruitment, could sensitize tumors to checkpoint blockade. We treated PyMT tumor-bearing mice with a combination of PD1 and CTLA4 blocking antibodies. While the blocking antibodies alone did not slow down tumor growth, consistent with previously reported lack of responses in this tumor model[44], PD1 and CTLA4 blockade significantly decreased tumor burden in iCoup mice, which already had smaller tumors than the control mice (Fig. 8A). Similarly in control KPC-tumor-bearing mice, PD1 and CTLA4 inhibition slightly though not significantly decreased tumor burden, but the checkpoint blockade in iCoup mice further inhibited tumor growth significantly (Fig. 8B). Taken together, these data suggest that COUP-TFII-driven endothelial reprogramming can sensitize tumors to immune checkpoint blockade in these models that are otherwise not responsive.

Given the enhanced recruitment of tumor antigen-specific T cells in iCoup mice (Fig. 5B), we reasoned that COUP-TFII-mediated endothelial reprogramming could also enhance the efficacy of adoptive T cell transfer therapy. We treated orthotopic PyMT-OVA tumor-bearing mice with a single injection of ex vivo-activated OT1 cells. OT1 transfer itself was not effective in restraining PyMT-OVA growth in control mice, but iCoup-driven endothelial reprogramming was significantly sensitized tumor responses to OT1-mediated inhibition (Fig. 8C). Similar sensitization was seen in orthotopic KP-OVA-bearing mice (Fig. 8D). Collectively, these data suggest that capillary-to-venule reprogramming not only enhances endogenous antitumor T cell responses, but also sensitizes tumors to immune checkpoint blockade and adoptive T cell transfer therapy.

## Discussion
In this study, we demonstrate that ectopic expression of COUP-TFII in tumor endothelium induces the molecular programs and functions of post-capillary venules, which play unique roles in regulating leukocyte trafficking. COUP-TFII-driven venular reprogramming promotes the recruitment of both endogenous antitumor T cells and adoptively transferred tumor antigen-specific T cells, thereby inhibiting tumor growth and amplifying the efficacy of immunotherapies.

Mechanistically, our data reveal that COUP-TFII-induced reversal of tumor immune evasion relies on T cells, correlates with increased effector T cell recruitment, and depends on vascular selectins and T cell CXCR3 to facilitate effector T cell homing to the tumor. It is worth noting that our pan-endothelial genetic manipulation approach affects not just the tumor blood vasculature but also lymphatic vessels and vessels at distant sites, potentially modulating the tumor immune responses further. The approach we explore here reprograms capillary endothelium into post-capillary venules for enhanced T cell recruitment. Notably, we did not observe de novo emergence of mature high endothelial venules (HEVs), the activation of which has been shown to subvert immune evasion[45-47]. Whether COUP-TFII-mediated venular programming could synergize with factors that support the differentiation of HEVs would further enhance antitumor immunity remains to be seen.

Surprisingly, the introduction of COUP-TFII expression in ECs selectively induced the infiltration of T cells but had little impact on myeloid cell infiltration. Some myeloid populations, such as TAMs, even slightly decreased in the orthotopic PyMT tumor model. The mosaic COUP-TFII induction experiments suggest that iCoup autonomously induces EC expression of CXCL10, a chemokine that attracts T cells expressing CXCR3, which also proved necessary for the enhanced T cell recruitment and tumor inhibition. Conversely, ectopic COUP-TFII repressed the expression of myeloid cell chemoattractants CCL2 and CCL5 (Fig. 3H)[38-41]. This differential regulation of venular chemokines may selectively sculpt the local immune niche. How COUP-TFII in ECs drives differential chemokine expression on the molecular level awaits elucidation.

This study supports the emerging concept that tumor endothelial cells in cold tumors are specialized not only for angiogenesis, but also for tumor immune evasion. The corollary of this hypothesis is that disruption of specialized EC phenotypes may reverse immune suppression. Our results identify programming TECs to venules as a promising avenue for immune therapy. Recent studies also identify metabolic and epigenetic programs of TECs that could be modulated to enhance tumor immunity[48,49]. These studies, combined with the unique accessibility of the blood vasculature, makes the endothelium a highly attractive therapeutic target. Techniques such as viral and lipid nanoparticle approaches have proven effective in delivering genes to blood vascular endothelial cells in mouse models[50,51]. Combining these approaches with endothelium-specific[52] or tumor EC-specific promoters will mitigate undesirable effects on bystander stromal or tumor cells.

Our study underscores the therapeutic potential of harnessing developmental programs to shape the specialization of tumor endothelial cells. The diversity among vascular endothelial cells, unveiled by the single-cell revolution, alongside insights from developmental biology, is uncovering genes and mechanisms responsible for the varied endothelial phenotypes in adult organs[53-55]. While we focused on tumor immunity, the pronounced effects of specialized EC programming suggest broader therapeutic applications, potentially benefiting patients with cardiovascular and autoimmune diseases.

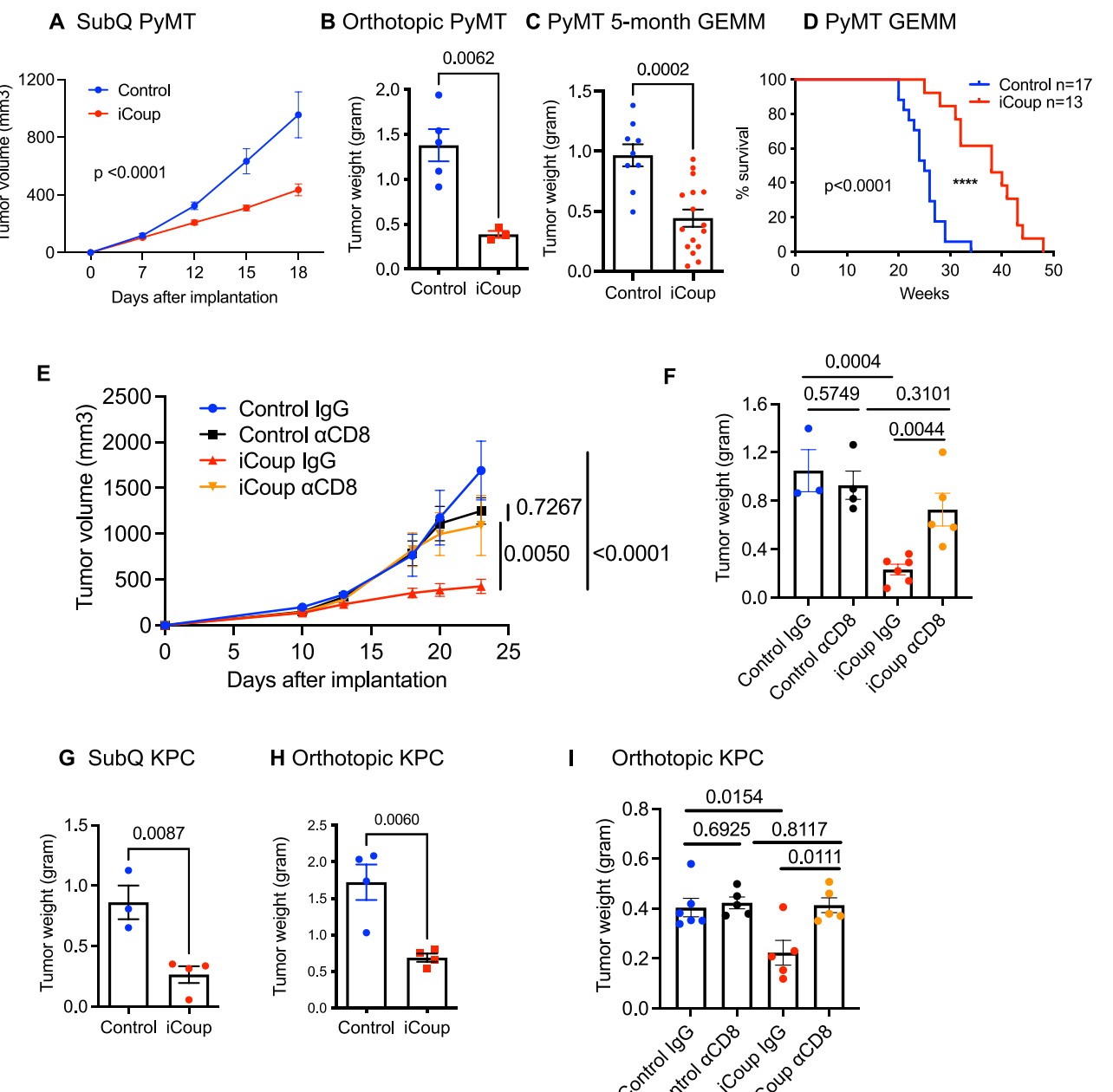

**Fig. 6 | COUP-TFII-driven endothelial reprogramming inhibits tumor growth in a T cell-dependent manner. A–C** Tumor volume or wet weight measurements of subcutaneous (subQ) (**A**), orthotopic (**B**), or autochthonous (**C**) PyMT tumors in iCoup and control mice. **A, B** is representative of more than five independent experiments. (**A** control: $n = 6$, iCoup: $n = 7$; **B** control: $n = 5$, iCoup: $n = 3$; **C** control: $n = 9$, iCoup: $n = 16$). **D** Survival curve of MMTV-PyMT-iCoup and control MMTV-PyMT mice. ****$p < 0.0001$ by log-rank (Mantel–Cox) test. (control: $n = 17$, iCoup: $n = 13$). **E, F** Tumor volume and wet weight measurements of orthotopic PyMT tumors in iCoup and control mice treated with CD8-depleting antibodies or control

IgGs. (control IgG: $n = 3$, control αCD8: $n = 4$, iCoup IgG: $n = 6$, iCoup αCD8: $n = 5$). **G, H** Tumor volume or wet weight measurements of subcutaneous (**A**) and orthotopic (**B**) KPC tumors in iCoup and control mice. ($n = 3$–4/group). (**G** control: $n = 3$, iCoup: $n = 4$; **H** $n = 4$/group). Representative of more than three independent experiments. **I** Wet weight measurements of orthotopic KPC tumors in iCoup and control mice treated with CD8-depleting antibodies. (control IgG: $n = 6$, control αCD8: $n = 5$, iCoup IgG: $n = 6$, iCoup αCD8: $n = 5$). Groups were compared by two-way ANOVA (**A, E**), unpaired two-tailed Student's $t$-test (**B, C, F–I**), or log-rank (Mantel–Cox) test (**D**). Error bars indicate s.e.m.

In conclusion, our study demonstrates the pivotal role of the vascular endothelium in tumor immune evasion. Targeted reprogramming with transcriptional programs that control physiologic EC specialization, as demonstrated here, presents a promising avenue to reverse immune evasion and activate antitumor immunity. These insights will help drive the development of tools and models for therapeutic gene delivery, leveraging vascular control of immune responses for improved cancer treatment.

## Methods

### Tumor models and antibody treatment

To establish orthotopic PyMT models, 50,000 or 100,000 PyMT tumor cells resuspended in 50 μL of Cultrex (Trevigen) were injected into the fourth mammary gland on the right side of age-matched mice, including Cdh5-Cre$^{ERT2}$; Rosa26$^{COUP-TFII}$ (homozygous for Rosa26$^{COUP-TFII}$) and control mice (Cdh5-Cre$^{ERT2}$ only or Rosa26$^{COUP-TFII}$ only). To establish KPC models, 20,000 or 50,000 KPC cells in 50 μL of Cultrex were

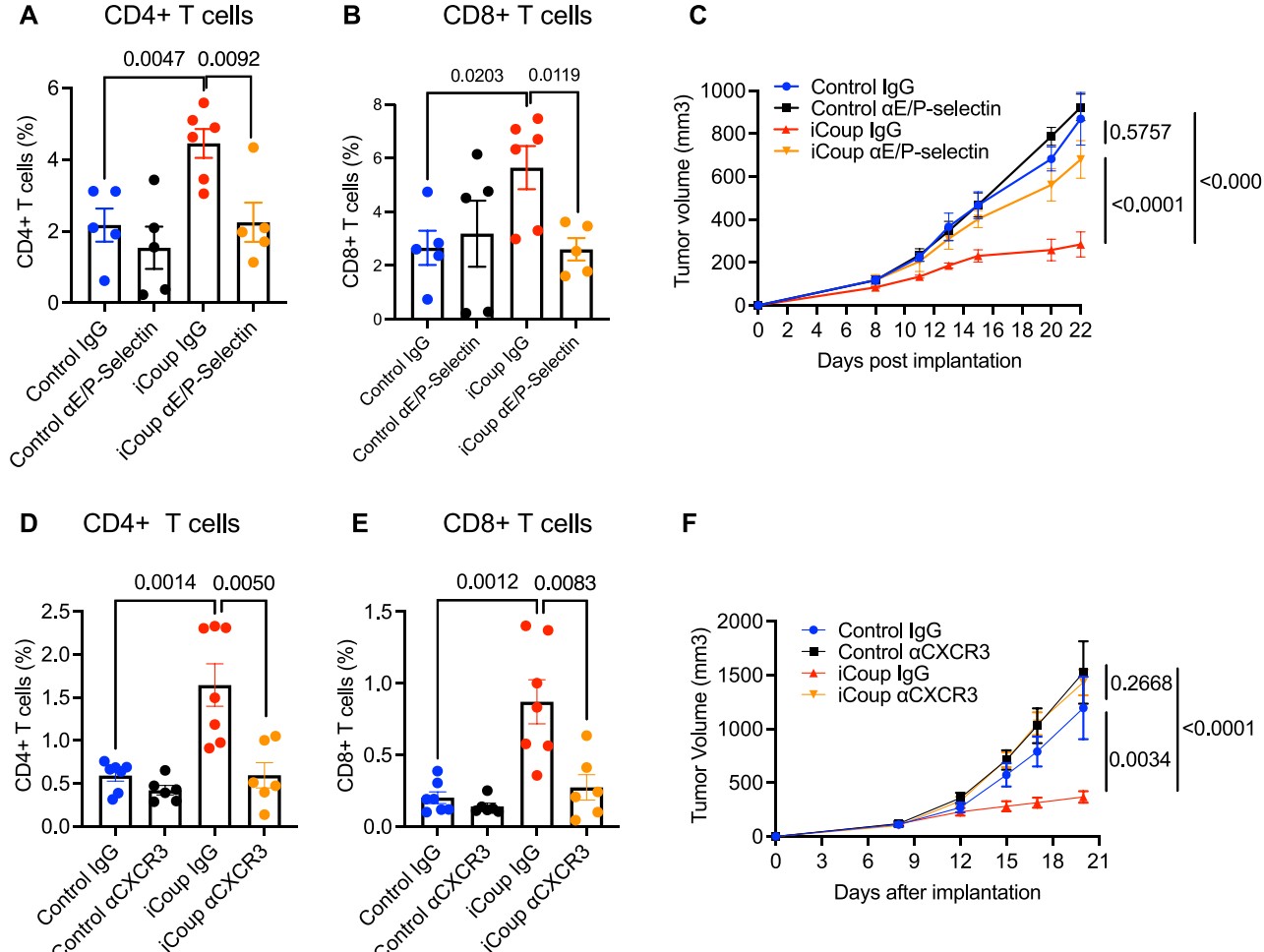

**Fig. 7 | Venule-mediated antitumor T cell recruitment requires COUP-TFII induction of E/P-selectins and CXCR3 signaling. A**, **B** Flow cytometric quantification of the frequency of tumor-infiltrating CD4+ or CD8 + T cells in subcutaneous PyMT-bearing iCoup or control mice, treated with E- and P-selectin blocking antibodies or control IgGs. **C** tumor volume measurement of mice in (**A**, **B**). **A**–**C** Control IgG: $n = 5$, Control αE/P-selectin: $n = 5$, iCoup IgG: $n = 6$, iCoup αE/P-selectin: $n = 5$. Representative of two independent experiments. **D**, **E** Flow cytometric quantification of tumor-infiltrating CD4+ and CD8 + T cells in sub-cutaneous PyMT-OVA-bearing iCoup or control mice, treated with CXCR3 blocking antibodies or control IgGs. ($n = 6$–7/group). **F** Tumor volume measurement of mice in (**D**, **E**). **D**–**F** Control IgG: $n = 7$, Control αCXCR3: $n = 6$, iCoup IgG: $n = 7$, iCoup αCXCR3: $n = 6$. Representative of three independent experiments. Groups were compared by unpaired two-tailed Student's $t$-test (**A**, **B**, **D**, **E**) or two-way ANOVA (**C**, **F**). Error bars indicate s.e.m.

injected orthotopically into the pancreas or subcutaneously into the left flank of the mice. Once tumors became palpable orthotopically or reached ~0.5 cm in width subcutaneously, mice were treated with two doses of 150 mg/kg tamoxifen. Concurrent with the first tamoxifen injection, where indicated, we also started treating mice intraperitoneally (i.p.) with antibodies against P-selectin (RB40.34, 200 mg), E-selectin (9A9, 200 mg), CD4 (GK1.5, 200 mg), CD8β (53-5.8, 200 mg), CTLA4 (9H10, 200 mg), CXCR3 (CXCR3-173, 10 mg/kg), PD1 (RMP1-14, 200 mg) or isotype controls, including Armenian hamster IgG, Syrian Hamster IgG, and rat IgG1 (HRPN), IgG2a (2A3), or IgG2b (LTF-2), twice a week. Most antibodies were purchased from BioXCell; P-selectin antibody was produced in house. Tumor volume was quantified as ½ × length × width². 

To modulate COUP-TFII expression in autochthonous tumors, we established MMTV-PyMT, Cdh5-Cre[ERT2], and LSL-COUP-TFII mice (MMTV-Coup for short). Mice were treated with two doses of tamoxifen at 150 mg/kg when they reached 3-month of age. One cohort of mice were sacrificed at 5-month of age for tumor weight measurement. A second cohort of mice were monitored for survival. Death was scored when (a) mice lost 15% of body weight, (b) tumors reached 1.5 cm in diameter, or (c) upon natural death. Log-rank (Mantel–Cox) test was performed for statistical analysis. All animal work was

approved by Institutional Animal Care and Use Committee at the Veterans Affairs Palo Alto Health Care System.

All mouse strains and tumor cell lines are under the C57BL6 background. For orthotopic and subcutaneous tumor models, age-matched mice within 3–6 months of age were used. A mix of male and female mice were used for the KPC and KPC-OVA models. Female mice were used for the PyMT and PyMT-OVA models. Mice were randomly selected for treatment groups.

**Tissue processing for flow cytometry and fluorescence-activated cell sorting (FACS)**

Tumor tissues were manually minced and transferred to the digestion buffer. Digestion buffer was prepared by adding collagenase D (Worthington Biochem) (400 U/mL for PyMT tumors or 500 U/mL for KPC tumors), 20 μg/mL DNase I (Sigma), and 2% fetal bovine serum (FBS), into Hanks Balanced Salt Solution (HBSS) (Thermo Fisher). Tumors were digested for 30 min at 37 °C with constant stirring, then quenched by ethylenediaminetetraacetic acid (EDTA), filtered through 40 μm Nylon mesh, pelleted by centrifugation (750 g for 5 min at 4 °C), and suspended in phosphate-buffered saline (PBS) for staining.

Blood was obtained through tail bleeding on live mice. Draining lymph nodes were isolated from sacrificed mice, strained against a 40-

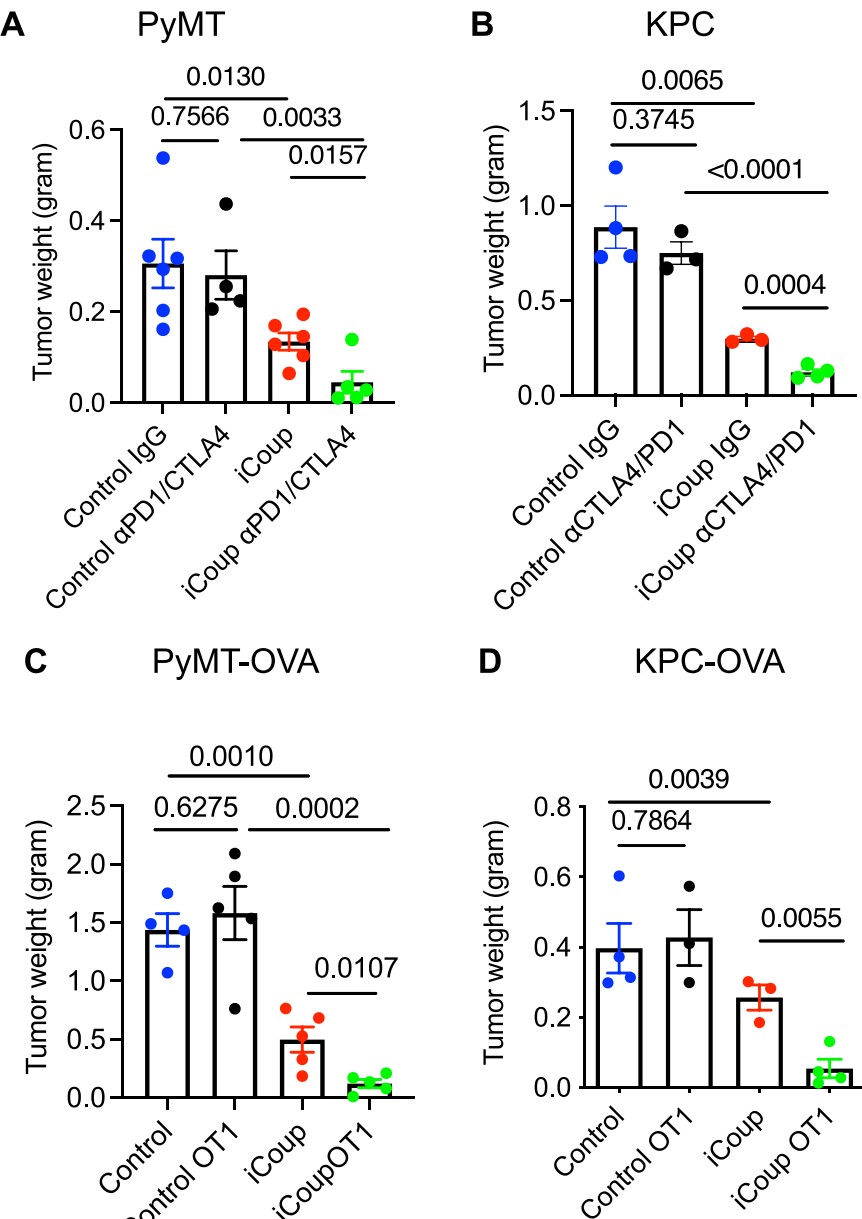

**Fig. 8 | COUP-TFII-driven venule programming sensitizes PyMT and KPC tumor responses to immunotherapies. A**, **B** Wet weight measurements of subcutaneous PyMT (**A**) and subcutaneous KPC tumors (**B**) in iCoup and control mice treated with αCTLA4 and αPD1 or control antibodies. (**A** control IgG: $n = 6$, control αPD1/CTLA4: $n = 4$, iCoup IgG: $n = 6$, iCoup αPD1/CTLA4: $n = 5$; **B** control IgG: $n = 4$, control αPD1/CTLA4: $n = 3$, iCoup IgG: $n = 3$, iCoup αPD1/CTLA4: $n = 4$). **C**, **D** Wet weight

measurements of orthotopic PyMT-OVA (**C**) and KPC-OVA (**D**) tumors in iCoup and control mice treated with ex vivo-activated OT1 cells. (**C** control: $n = 4$, control OT1: $n = 5$, iCoup: $n = 5$, iCoup OT1: $n = 5$; **B** control: $n = 4$, control OT1: $n = 3$, iCoup: $n = 3$, iCoup OT1: $n = 4$). Groups were compared by unpaired two-tailed Student's $t$-test unless. Error bars indicate s.e.m. Representative of two independent experiments. $n = 3-6$/group.

μm Nylon mesh, and rinsed with PBS. Blood and LNs were incubated in red blood cell lysis buffer (BioLegend) for 3 min and quenched in staining buffer (PBS with 1% FBS).

Cell suspensions were incubated in PBS with anti-mouse CD16/CD32 antibodies (at 1/200 dilution) (eBioscience) and Zombie NIR™ Fixable Viability dye (at 1/500 dilution) (BioLegend) for 10 min, pelleted by centrifugation, and then incubated with 100 μL of fluorophore-conjugated antibodies at pre-optimized dilutions for 20 min on ice, and washed with staining buffer (PBS with 1% FBS). Single cell suspension was incubated with fluorophore-conjugated antibodies, washed in staining buffer, fixed in 4% formaldehyde for 30 min on ice, and resuspended in staining buffer. Data were acquired on LSR-Fortessa (BD Biosciences) and analyzed using the FlowJo software.

To assess hypoxia, tumor-bearing mice were injected with pimonidazole intraperitoneally at a dose of 60 mg/kg. Mice were euthanized 1 h later. Single-cell suspensions were processed as above. After surface antibody staining, cells were permeabilized with eBioscience™ FoxP3/Transcription Factor Staining Buffer Set (Thermo Fisher), and stained with fluorophore-conjugated antibodies for PIMO adducts for 20 min at room temperature.

To assess T cell proliferation, mice were injected with 4 mg of 5-bromo-2-deoxyuridine (BrdU) 4 h prior to euthanasia. The BD Pharmingen™ FITC BrdU Flow Kit (BD Biosciences) was used for intracellular BrdU staining.

To sort BECs for single-cell RNA sequencing, single cell suspensions from digested tumors were treated with red blood cell lysis

buffer, washed in PBS, and incubated with biotinylated antibodies against lineage markers (CD45, EpCAM, Ter119, podoplanin, and Annexin V) for 20 min. Cells were then pelleted and incubated with streptavidin microbeads (Miltenyi Biotec), and lineage+ cells were depleted using the MACS LS columns (Miltenyi Biotec). Cells were then stained with fluorescently labeled antibodies, and BECs were sorted on FACSAria 2 or Aria 3 (BD Biosciences) into 1 mL of FBS at 4 °C, and immediately prepared for 10x Chromium v3 sequencing. To sort BECs for QPCR analyses, cells were sorted on Aria2 or Aria 3 directly into E.Z.N.A. lysis buffer (Omega Bio-tek). Sorting purity was confirmed to be >95% before proceeding.

## Mosaic COUP-TFII induction

PyMT or KPC tumor cells were orthotopically injected into Cdh5-Cre$^{ERT2}$, Rosa26$^{COUP-TFII}$, Rosa26$^{tdTomato}$ (heterozygous) mice. Mice were injected with a single dose of tamoxifen at 7.5 mg/kg of body weight on Day 5 after tumor implantation. When tumors reached 1.5 cm in diameter, ~21 days after tumor implantation, mice were sacrificed and tumors were dissociated as described above. Single cell suspension was incubated first with biotinylated antibodies against EpCAM, CD45, and podoplanin, and then with streptavidin microbeads (Miltenyi Biotec) to deplete positive cells. Endothelial cell-enriched portions were incubated with CD16/CD32 and ZOMBIE NIR™ Fixable Viability dye in PBS, and then stained with fluorescently-conjugated streptavidin and antibodies against CD31 and lineage markers as described above. tdTomato + and (−) ECs were then sorted by BD Aria2 or Aria 3 into 700 µL of RNA lysis buffer (E.Z.N.A.) for subsequent QPCR analyses.

## RNA isolation and quantitative PCR (Q-PCR)

RNA was isolated using E.Z.N.A. Total RNA kit (Omega Bio-tek). cDNA was synthesized using qScript cDNA SuperMix (Quantabio) following the manufacturer's instructions. Q-PCR was performed using SYBR Green Master mix (Thermo Fisher). Primers were either designed using Primer3 or referenced from PrimerBank and synthesized by Integrated DNA Technologies. Primer sequences are listed in Supplementary Table 1. For endothelial cells sorted from mouse tumors, cDNA was pre-amplified with SsoAdvanced™ PreAmp Supermix (BioRad) following the manufacturer's protocol before proceeding to Q-PCR.

## Adoptive transfer of ex vivo-activated OT1 cells

The spleens and lymph nodes (cervical, brachial, axillary, inguinal, hepatic, portal, and mesenteric) were harvested from 6- to 12-week-old OT1 mice, lysed the red blood cells, and filtered through 40-µm Nylon mesh. Cells were then activated in Roswell Park Memorial Institute (RPMI)−1640 (Gibco) containing 10% FBS, 50 µM β-mercaptoethanol, 1x Pen-Strep (Gibco), 0.5 µg/mL SIINFEKL peptide (Sigma-Aldrich) and 10 ng/mL recombinant murine IL-2 (Peprotech). Fresh media with SIINFEKL peptide and IL-2 was added every 2 days. On the day of transfer, cells were enriched using CD8α + T cell isolation kit (Miltenyi Biotec), incubated with CFSE in serum-free RPMI-1640 at 37 C for 10 min, washed in PBS, and resuspended in 2−10 × 10$^6$ cells per 200 uL of PBS, and immediately injected intravenously into recipient mice.

## Leukocyte homing assays

For the first setup, PyMT tumors were established in Cdh5-Cre$^{ERT2}$; Rosa26$^{COUP-TFII}$, and control mice. When tumors reached ~0.5 cm in diameter, we obtained draining lymph nodes from a cohort of mice, filtered through 40-µm Nylon mesh, and lysed red blood cells. We incubated cells in RPMI-1640 media (less than 5 × 10$^7$ cells per 8 mL of media) containing CFSE (Invitrogen) at 625 nM for 10 min at 37 °C. Cells were then quenched with FBS, washed twice with RMPI, and pelleted by centrifugation. Equal numbers of CFSE-labeled cells (up to 1 × 10$^7$ cells per mouse in 200 µL of PBS) were injected intravenously (i.v.) into tumor-bearing mice. Mice were sacrificed 16 h later, and tumor tissues were processed for flow cytometric analyses as discussed above.

For the second setup, PyMT-OVA or KPC-OVA tumor cells were implanted orthotopically into Cdh5-Cre$^{ERT2}$; Rosa26$^{COUP-TFII}$, and control mice. When tumors reached ~0.5 cm in diameter, we injected mice with ex vivo-activated, CFSE-labeled OT1 cells intravenously. To activate OT1 cells ex vivo, we obtained spleen and lymph nodes (brachial, auxiliary, or inguinal for PyMT-OVA recipients, or hepatic, portal, and mesenteric for KPC-OVA recipients); tissues were filtered through 40 µm Nylon mesh, and then incubated in RPMI-1640 containing 10 ng/mL of recombinant murine IL-2 (Peprotech), 0.5 µg/mL of SIINFEKL peptide (Sigma-Aldrich), and 50 µM of beta-mercaptoethanol. We obtained tumor tissues for flow cytometry 16 h later.

For the third setup, we isolated bone marrow cells from the femur, tibia, and humerus of female Rosa26$^{mTmG}$ mice, lysed red blood cells, and injected i.v. into PyMT tumor-bearing mice. Homing was also assessed 16 h after injection.

## Transwell migration assay

The cell migration assay was performed using transwell migration chambers (3.0-µm pore size, Sigma-Aldrich). PyMT tumor cells were plated at a density of 5 × 10$^4$ cells/well in 24-well plates. Upon adhesion to the plate ~6 h later, tumor cell culture was switched to 1 mL of FBS-free DMEM with 1X penicillin/streptomycin; cells were cultured until reaching 90% confluency ~2 days later. bEND3-control and bEND3-COUP-TFII cells were plated into transwell inserts at a density of 2 × 10$^5$ cells/insert to reach 100% confluency, and allowed 6 hours to adhere. The mouse spleens were then obtained from 6-to-8-week-old C57BL6 mice, filtered through 40-µm Nylon mesh, processed for red blood cell lysis, added into the inserts at a density of 1 × 10$^6$ cells/insert, and transwell inserts were added on top of the lower chambers containing tumor cells and conditioned media to induce splenocyte migration. Twelve hours later, cells from the lower chamber were harvested and stained with fluorophore-conjugated antibodies for lineage markers and TCRβ; CountBright counting beads (Invitrogen) were added to each sample to calculate the number of transmigrated cells per well.

## Single-cell RNA sequencing

PyMT or KPC tumor cells (100,000) were implanted in 6-week-old female C57BL6 mice. Ten days after implantation, tumors were digested as described above, and endothelial cells were sorted and processed for scRNAseq using Chromium Single Cell 3' Library and Gel Bead Kit v2 (10x Genomics) according to manufacturer guidelines. Libraries were sequenced on NextSeq500 (Illumina) at Stanford Functional Genomics Facility. Cell Ranger (v3.0, 10x Genomics) was used to align reads to the mm10 reference genome and perform quality control and cell assignment. Normalization and log transformation of cell X gene count data were computed using the quickCluster and computeSumFactors functions from the scran package and the logNormCounts from the scuttle package. Cell identities were determined by maximal correlation with average expression profiles from previously annotated lymph node blood endothelial cell datasets[19], and were visualized with uniform manifold approximation and projection (UMAP) computed using the umap package in R with default parameters. Contaminating cells were removed by supervised gating based on QC parameters and canonical marker expression. To recover gene–gene relationships that are lost due to dropouts, missing gene expression data from log-normalized count data was imputed using the MAGIC (Markov Affinity-based Graph Imputation of Cells) algorithm with optimized parameters ($t = 2$, $k = 9$, $ka = 3$)[56]. Imputed data were used for visualization of single-cell gene expression in dot plots.

## Immunofluorescence staining

Isolated tumor tissues were immediately embedded in optimum cutting temperature (OCT) compound on dry ice. Alternatively, tissues were fixed in 4% formaldehyde for 2 h at room temperature, incubated

in 30% sucrose overnight, and then embedded in OCT compound and frozen on dry ice. Sections of 10-μm thickness were incubated with 5% normal goat serum (Jackson Immunoresearch) in phosphate-buffered saline for 1 h at room temperature. Sections were then treated with primary rabbit or rat antibodies recognizing indicated mouse epitopes suspended in PBS at 4 °C overnight. Slides were then washed in PBS with 0.1% Triton-X, with gentle rocking for three washes for 10 min each, followed by incubation with goat secondary antibodies against rat or rabbit IgG, or with fluorophore-conjugated antibodies for 1 h at room temperature (Thermo Fisher). Slides were again washed in PBS with 0.1% Triton-X for 3 × 10 min, and mounted in DAPI-containing Fluoromount mounting media (Southern Biotech). Images of fluorescence staining were acquired on an Apotome microscopy (Nikon).

## Statistical analysis
Statistical analyses were performed using the GraphPad Prism software (v10). Specific statistical tests for experiments, including the two-tailed student's $t$-test and log-rank (Mantel–cox) test, are listed in the figure legends. All values are expressed as ±SEM. $P$ values less than 0.05 were considered significant.

## Reporting summary
Further information on research design is available in the Nature Portfolio Reporting Summary linked to this article.

## Data availability
All data supporting the findings of this study are available within the article, Supplementary Information, or Source Data files. Raw single-cell RNAseq data have been deposited in the Gene Expression Omnibus (GEO) database under accession code: GSE281284. Source data are provided with this paper.

## Code availability
Code used in this study, including a customized implementation of the MAGIC algorithm, are available as part of the magicBatch package [https://github.com/kbrulois/magicBatch].

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

## Acknowledgements

The authors acknowledge support from NIH R01 grants CA228019 and AI130471 (E.C.B.), TRDRP grant T33IR6609 (E.C.B.), Cancer Research Institute Irvington Postdoctoral Fellowship (Y.Z.), NIH grants F32 CA200103 (K.F.B.), T32 grants AI07290 and HL098049 (T.T.D.), and fellowships from American Heart Association and Stanford Cardiovascular Institute (T.T.D.). Sequencing was performed at Stanford Functional Genomics Facility with instrumentation funded by NIH S10OD025212 and 1S10OD021763. The PyMT and KPC cell lines were kind gifts from David G. DeNardo. The authors would like to thank all members of the Butcher Laboratory for input on this work.

## Author contributions

Conceptualization: Y.Z. and E.C.B. Investigation: Y.Z. and T.T.D. Methodology: Y.Z. and K.F.B. Analysis: Y.Z., K.F.B., and J.P. Manuscript: Y.Z. and E.C.B. Supervision: Y.Z. and E.C.B. Funding acquisition: Y.Z., K.F.B., T.T.D., and E.C.B.

## Competing interests

The authors declare no competing interests.
