## [Transparent Peer Review file · Nature Communications]

COUP-TFII mediated reprogramming of the vascular endothelium abrogates tumor immune evasion

Corresponding Author: Dr Yu Zhu

Version 0:

Reviewer comments:

Reviewer #1

(Remarks to the Author)

COUP-TFII is a master transcription regulator of venule differentiation, and the authors observed that ectopic expression of COUP-TFII (iCOUP) increases the expression of cell adhesion molecules and selectins in tumor endothelium. They suggest this as evidence of converting capillary vascular ECs into post-capillary venules which are more permissive to immune cell entry; especially anti-tumor immune cells such as T-cells. Accordingly, iCOUP mice have increased effector T cell infiltration in the tumor. As a result, iCOUP mice have better tumor inhibition and this inhibitory effect is dependent on CD8 T cells and the expression of selectins on the tumor endothelium. This is a very nice body of work using a series of GEM models and tumor allografts that shows how modifying vascular phenotypes can globally impact anti-tumor immunity.

Major points (in no particular order)

The authors did not discuss the robustness of experiments throughout the manuscript. In many figures the authors showed pooled data from independent experiments but did not mention how many animals/samples were included in the figures. For example, the number of data points from large tumors varies in Figure 1 D-F.

In Figure 1 the authors did not observe a significant difference in BEC percentages between smaller and larger tumors, but larger tumors have less T cell infiltration. Do the authors expect to see a downregulation of COUP-TFII expression in BECs from larger tumors?

All immune cell population from flow cytometry were presented as a percentage of total cells from tumor, which is dependent on tumor size. Normalizing the number of immune cells per mg of tumor can be used in together with the percentages. It is also curious the authors chose to evaluate these data by tumor size rather than by time.

The authors showed ectopic COUP-TFII expression can lead to capillary to venule reprogramming by looking at the expression of venule-specific genes. I am curious to see if this change in the expression of venule-specific genes also leads to changes in the morphology of the vasculature.

The authors induced COUP-TFII upon tamoxifen induction, but the authors did not show the overexpression efficiency of COUP-TFII in mice in terms of Nr2f2 expression.

Increased CD8 T cell infiltration was observed in MMTV-GEMM iCOUP mice (Figure 4L), whereas 4A-K were from orthotopic tumors. The authors should also include IF images from orthotopic model. I am also curious to see if those T cells are in close proximity with ICAM1+/VCAM1+/P-selectin+ venules. The data from Fig 4L are also not robust and have not been quantified (how many mice, tissue sections, etc?).

Do iCOUP mice show immune cell infiltration in other tissue/organ microenvironment; i.e. not just tumors?

It is confusing why the OT1 transfer in PyMT mice was not effective. Shouldn't this generate robust anti-tumor immune cells due the model antigen OVA?

Figure 6I – the authors should include control anti-CD4/CD8 for better comparison in KPC model. Also, the authors used a combination of anti-CD4 and CD8 instead of anti-CD8 alone when treating PyMT tumors. Can the authors provide a rationale why they are not using the same T cell depleting strategy between the two experiments?

The authors claimed the increase in T cell infiltration in iCOUP mice is not due to increased T cell availability in the circulation. Did the authors harvest the draining lymph nodes to see if iCOUP increases the T cell population in the lymph nodes?

An increase in tumor infiltrating T cells can result from both increased homing and increased T cell proliferation in tumor site. The authors did a series of experiments that suggested increased T cell homing, but no experiment was done to account for the possibility of increased T cell proliferation.

Not clear how robust data are in Fig 7. Is this a single experiment with only n=5 mice per group? This may be underpowered.

The annotation of the scRNAseq data fig S1 shows a surprising absence for lymphatic ECs. Were these filtered out of the analysis?

Minor

Fig S3D is presented out of order in the manuscript; i.e. the callouts for figures usually should appear sequentially.

Reviewer #2

(Remarks to the Author)

The manuscript by Zhu et al. addresses an important question regarding the role of endothelial cell specialization in anti-tumor immune responses. The authors propose that the immunosuppressive tumor microenvironment results from a paucity of tumoral venous endothelial cells. They further show that endothelial cell specific overexpression of COUP TFII is sufficient to promote venous endothelial cell expansion, increase infiltration of T lymphocytes, and decrease growth of tumors in two different models. Overall, the work is of interest and importance for the field. I have the following comments:

1. The COUP TFII overexpression model needs to be characterized in more details.
 - What are the level of COUP TFII overexpression in different tumor endothelial compartments, i.e. capillaries, venous endothelial cells and arterial endothelial cells?
 - Is such expression heterogeneous or uniform?
 - Are newly generated venules located intratumorally or peritumorally?
 - What is the impact of COUP-TFII overexpression on vascular density, pericyte recruitment and tumor hypoxia?
2. The impact on capillary endothelial cells is well described, what was the effect of COUP TFII overexpression on arterial endothelial cells?
3. Do the authors observe emergence of HEVs in their models?
4. In many experiments the authors analyse three tumors per condition. Although the results are statistically significant, such low number makes difficult to judge the robustness of the observed effect.

Minor comment

1. error bars should be shown as standar deviations, not SEM

Reviewer #3

(Remarks to the Author)

The manuscript entitled 'Tumor immune evasion abrogated by COUP-TFII reprogramming of the vasculature' by Zhu et al, presents evidence that molecular reprogramming of endothelium by COUP-TFII expression supports anti-tumor T cell responses. The authors demonstrate that endothelial expression of COUP-TFII, a transcription factor that drives venous development, induces postcapillary venules in tumors, which facilitates recruitment of T cells into the tumor and suppresses tumor growth, and improves responsiveness to immunotherapy. Current immunotherapy is successful only in a subset of cancer patients, and strategies to enhance anti-tumor immunity in immunologically "cold tumors" are warranted. Here the authors provide an important contribution to this area.

The manuscript presents strong in vivo data that promotion of post-capillary venule phenotype in ECs rendered by COUP-TFII facilitates T cell migration into tumors. These findings are replicated in several models, lending credibility. Nevertheless, some conclusions deserve better/additional supporting evidence to make author's reasoning stronger. The specific points are summarized below.

Major comments

1. The authors present in vivo evidence that mice expressing COUP-TFII in endothelial cells (iCoup mice) facilitate enhanced T cell uptake into tumors. However, given the complexity of the tumor microenvironment (TME), these proofs are indirect. An assay directly demonstrating the functional role of endothelial cells in this process, e.g. in isolated endothelial cells in vitro, is required.

2. Increased ICAM1 and VCAM1 expression in COUP-TFII expressing tumor endothelial cells is shown by flow cytometry measurement. While quantitative, this approach lacks spatial resolution. Is the vascular expression of ICAM1 and VCAM1 uniform in the TME or are there specific areas with increased local expression of these proteins in WT vs iCoup mice?
3. The results of a homing assays depicted in Figure 5A, 5B, and 5C, are based on very few cells. It is important that these experiments are reproduced across at least three independent cohorts of mice to ensure reliability of the results.
4. In Figure 5E, endothelial remodeling is shown to influence the population of B cells in the blood. However, data on the B cell population within tumors is not shown. Are B cells changed also in the tumor? Is this functionally important?
5. Figure 1L shows a change in the ratio of T cell infiltration per P-selectin positive endothelial cells between small and large tumors, suggesting a potential P-selectin independent effect on T cell infiltration. However, an unchanged ratio would be anticipated if P-selectin positive endothelial cells were solely responsible for T cell infiltration. Can the authors provide a better explanation of this figure? Accordingly, please correct the conclusions drawn from this data in the text (lines 90-91).
6. For all figures, the number of independent experiments and the number of mice used in each experiment must be indicated in the legend.
7. The "Materials and Methods" section is incomplete. Please add information for the following experiments:
 - a. Immunofluorescence (Figure 4L) and FACS analysis of blood samples (Figure 5D and E).
 - b. Detailed information regarding the data analysis of single-cell RNA sequencing, including specification of the algorithms and settings used. Also, the authors must upload their sequencing data of the scRNAseq to a dedicated repository (ArrayExpress), and provide an accession number in the methods section.
 - c. Detailed information for the mouse models, such as genetic background, sex and age distribution.
 - d. Method section for statistical analysis is missing.
8. The "Discussion" section should be extended with a more thorough discussion of results in the context of existing literature, such as by Gabrielle Bergers and Jean-Philippe Girard.

Minor comments

1. In Figure 2G and text line 121, the name of the capillary marker Podocalyxin is misspelled. Please correct.
2. Figure 4 M-R is erroneously referenced as Figure 5 (lines 176-177). Additionally, treatment compound used in Figure 4 M-R should be specified.
3. In Figure 4L, scale bars are missing.
4. In all the figures containing gating strategies (5D, S2A-C), authors should ensure that the text describing gate populations does not overlap with other texts in the figure.
5. The legend for Figure 5C is incomplete. Please correct.
6. Still in Figure 5C, the authors should indicate in the graph that the population of CD8+ T cells is shown as was done in Figure 5A and 5B.
7. In Figure 6A please use cubic millimeters (instead of centimeters) to maintain consistency among the figures showing tumor volume.
8. In Figure 6E, statistics are missing. Please adapt the related text and conclusions if there was no significant difference,.
9. A control group treated with depleting antibodies should be included in Figure 6I. Also, please explain why different depleting strategies were used in Figure 5F and 5I.
10. Authors should use the same group names across related figures, e.g. Figure 7A -7C and 7D – 7F. Also, names/labels of graph axes and conditions should be used consistently throughout the manuscript.
11. In Figure 7C and 7F it is unclear which treatment groups are being compared for statistical evaluation. Please re-align the vertical lines.
12. Can the authors explain why both CD4+ (Figure 7A) and CD8+ (Figure 7B) T cells are shown in some experiments, while only the entire population of T cells is shown in others (Figure 7D)? If possible, please show the same populations to facilitate data interpretation.
13. The text (lines 83-84) and figure legend associated with Supplementary Figure S1A do not match the displayed graphs. Please correct.
14. In Supplementary Figure S1 E and F please add the correlation coefficients.

Version 1:

Reviewer comments:

Reviewer #1

(Remarks to the Author)

A very nice study. This reviewer's comments have been adequately addressed.

Minor comments:

on line 32 do you mean "tumor nests" rather than "nets"?

on line 246 should be "...recruited into the tumor tissue."

on line 290 is "CD8beta-depleting" correct?

in fig S3, "tumor nests" or "tumor nets"?

Reviewer #3

(Remarks to the Author)

The authors addressed most of the comments. However, a few points remain that might have been missed and need to be addressed.

Most importantly, while the Material and Methods section has been updated, detailed information on the analysis of scRNA-seq data, including the specific algorithms and settings used in the analysis, is missing. Without this information the analysis cannot be reproduced. For the same reason, as requested previously, the authors need to upload the scRNA-seq data to a suitable repository (such as GEO, ArrayExpress) and include the accession number in the methods section.

Minor unsolved issues are:

1. The authors indicated that the legend for graph S1A had been revised, but it seems to be unchanged. Please update the legend to ensure that it clearly describes all three figures.
2. In the legend for Figure 5C, the authors state that "Quantification indicates the frequency of homed donor CD8+ T cells as % of total cells dissociated from the tumor". However, the figure includes four graphs showing the quantification of different cell types.
3. The authors were asked to use consistent group names across related figures, such as Figures 7A-7C and 7D-7F, but Figure 7F was missed in this process.
4. For Figure 8, please indicate the number of independent experiments and the number of mice used in each experiment.

Version 2:

Reviewer comments:

Reviewer #3

(Remarks to the Author)

All my remaining comments have been addressed. I would like to congratulate the authors to a nice study.

To all reviewers:

Thank you for your efforts in providing a thorough review of our paper. Here we provide a revised manuscript addressing your concerns. To aid the review, the responses are color coded. **Black** is the reviewer's comments and **Blue** is our response. Major changes in the manuscripts are highlighted with a bar on the left side of the text.

Thank you for taking the time to review our revision.

Reviewer #1 (Remarks to the Author): with expertise in cancer immunology, tumor vasculature

COUP-TFII is a master transcription regulator of venule differentiation, and the authors observed that ectopic expression of COUP-TFII (iCOUP) increases the expression of cell adhesion molecules and selectins in tumor endothelium. They suggest this as evidence of converting capillary vascular ECs into post-capillary venules which are more permissive to immune cell entry; especially anti-tumor immune cells such as T-cells. Accordingly, iCOUP mice have increased effector T cell infiltration in the tumor. As a result, iCOUP mice have better tumor inhibition and this inhibitory effect is dependent on CD8 T cells and the expression of selectins on the tumor endothelium. This is a very nice body of work using a series of GEM models and tumor allografts that shows how modifying vascular phenotypes can globally impact anti-tumor immunity.

Response: We thank this reviewer for the enthusiasm and positive comments.

Major points (in no particular order)

1. The authors did not discuss the robustness of experiments throughout the manuscript. In many figures the authors showed pooled data from independent experiments but did not mention how many animals/samples were included in the figures. For example, the number of data points from large tumors varies in Figure 1 D-F.

Response: Thank you for pointing this out. We have addressed the robustness of experiments throughout the manuscript. We have substantially increased the sample sizes in many figures. Figures and legends have been updated to include detailed information on the sample sizes and number of independent repeats. We updated new data for Figure 1 including a larger sample size with a consistent number of data points across the panels.

2. In Figure 1 the authors did not observe a significant difference in BEC percentages between smaller and larger tumors, but larger tumors have less T cell infiltration. Do the authors expect to see a downregulation of COUP-TFII expression in BECs from larger tumors?

Response: The reviewer is correct. While the abundance of BECs was not significantly different between smaller and larger tumors, we did see a significant downregulation of P-selectin+ post-capillary venules, i.e. the predominant BEC subset that expresses COUP-TFII, suggesting an overall reduction in the abundance of ECs capable of recruiting lymphocytes.

Reviewer 1 Point 1 and Point 3

- All immune cell population from flow cytometry were presented as a percentage of total cells from tumor, which is dependent on tumor size. Normalizing the number of immune cells per mg of tumor can be used in together with the percentages. It is also curious the authors chose to evaluate these data by tumor size rather than by time.

Response: We have added new data comparing the number of immune cells per milligram of tumor, which is consistent with the comparisons based on the percentages. We opted to evaluate these data by tumor size instead of time to minimize technical variations. This would allow us to process and analyze as many tumors as possible in a single setting, reducing potential batch-to-batch variations.

- The authors showed ectopic COUP-TFII expression can lead to capillary to venule reprogramming by looking at the expression of venule-specific genes. I am curious to see if this change in the expression of venule-specific genes also leads to changes in the morphology of the vasculature.

Response: We have added new immunofluorescence images to show morphological features vasculature. Furthermore, we quantified the CD31+ covered area and assessed the pericyte coverage on the vessels. In addition, we used MAdCAM1 as a marker for newly generated venular ECs (as shown in Figures 2G and 2M, MAdCAM1 is expressed by BECs in iCoup but not control tumors). We show MAdCAM1+ ECs both in the keratin+ area and outside the keratin+ area, suggesting that the venules are both in the tumor nets and in the stroma. These figures are included in **Figures 2C, S3A, S3B**.

Review 1 Point 4

5. The authors induced COUP-TFII upon tamoxifen induction, but the authors did not show the overexpression efficiency of COUP-TFII in mice in terms of Nr2f2 expression.

Response: Using intracellular flow cytometry we show the overexpression of COUP-TFII on the protein level in tumor BECs of iCoup mice. By immunofluorescence imaging we also confirm the high penetrance of COUP-TFII specificity in endothelial cells in the iCoup model. These data are included in **Figures 2A-C**.

Reviewer 1 Point 5

6. Increased CD8 T cell infiltration was observed in MMTV-GEMM iCoup mice (Figure 4L), whereas 4A-K were from orthotopic tumors. The authors should also include IF images from orthotopic model. I am also curious to see if those T cells are in close proximity with ICAM1+/VCAM1+/P-selectin+ venules. The data from Fig 4L are also not robust and have not been quantified (how many mice, tissue sections, etc?).

Response: We performed statistical analyses to quantify the abundance of CD8+ T cells based on immunofluorescence imaging, which shows patterns consistent with the quantification based on flow cytometry (**Figures 4L-N**). We also included images of orthotopic tumor models. Most infiltrating CD8+ T cells are indeed in proximity to the vessels but have extravasated out of the blood vessels and infiltrated into the tumor cell nests, as demonstrated by the proximity between CD8+ cells and keratin+ cells.

Reviewer 1 Point 6

7. Do iCOUP mice show immune cell infiltration in other tissue/organ microenvironment; i.e. not just tumors?

Response: COUP-TFII overexpression in different tissue beds has varied impacts on the vascular endothelium and immune infiltration in different tissues. We assessed ECs and immune infiltration in the lung, liver, and kidney, two weeks after tamoxifen induction which follows the timeline of iCoup induction in our tumor settings. In the lung, we saw a downregulation of P-selectin upon iCoup induction. This correlates to slightly decreased frequencies of tissue T cells but an increased myeloid abundance. In the liver we did not observe extensive changes in ECs or immune infiltration, with the exception for slightly increased ICAM1 expression in BECs. In the kidney we did not see venular marker changes, but T cell abundance was decreased.

As we show in our previously published study, the impact of iCoup depends on the tissue of origin and the local endothelial properties. The impact of iCoup on the local immune infiltration is complex and the mechanisms require further dissection. These data are added to **Figures S4B-G**.

Reviewer 1 Point 7

8. It is confusing why the OT1 transfer in PyMT mice was not effective. Shouldn't this generate robust anti-tumor immune cells due the model antigen OVA?

Response: While activated OT1 cells can potently kill PyMT-OVA and KPC-OVA cells *in vitro*, they did not induce efficient anti-tumor responses *in vivo*. As suggested by the OT1 homing assay results, *ex vivo*-activated OT1 cells do not home efficiently to the tumor sites. We reason that the tumor microenvironment promotes immune evasion at least in part through the downregulation of lymphocyte recruitment programs at post-capillary venules.

9. Figure 6I – the authors should include control anti-CD4/CD8 for better comparison in KPC model. Also, the authors used a combination of anti-CD4 and CD8 instead of anti-CD8 alone when treating PyMT tumors. Can the authors provide a rationale why they are not using the same T cell depleting strategy between the two experiments?

Response: To keep consistency with the other figures, we included KPC tumor burden analyses upon the depletion of CD8+ T cells alone, which mirrors the effect of dual CD4/CD8 depletion and highlight the importance of CD8+ T cells in the anti-tumor effect mediated by iCoup. The data are now included in **Figure 6I**.

10. The authors claimed the increase in T cell infiltration in iCoup mice is not due to increased T cell availability in the circulation. Did the authors harvest the draining lymph nodes to see if iCoup increases the T cell population in the lymph nodes?

Response: We analyzed the composition of T cell subsets in the draining lymph nodes. The abundance of effector, memory, and naïve T cells was not significantly altered by iCoup. The data are now added to **Figures 5F-T**. However, we would like to emphasize that we do not rule out the possibility that alterations in the lymph nodes could contribute to the anti-tumor phenotypes we saw in iCoup mice, and we addressed this in the Discussion.

Reviewer 1 Point 10

11. An increase in tumor infiltrating T cells can result from both increased homing and increased T cell proliferation in tumor site. The authors did a series of experiments that suggested increased T cell homing, but no experiment was done to account for the possibility of increased T cell proliferation.

Response: Thank you for this suggestion. We added new data assessing BrdU incorporation, which suggest that the local T cell proliferation was not significantly changed in iCoup mice. These data are

now included in **Figures 40-P**.

Reviewer 1 Point 11

12. Not clear how robust data are in Fig 7. Is this a single experiment with only n=5 mice per group? This may be underpowered.

Response: These data have been robustly reproduced by independent experiments. We include newly repeated data with larger sample sizes.

13. The annotation of the scRNAseq data fig S1 shows a surprising absence for lymphatic ECs. Were these filtered out of the analysis?

Response: Yes, we focused our research on CD31+podoplanin-negative BECs and did not sort lymphatic ECs for this analysis.

Minor

14. Fig S3D is presented out of order in the manuscript; i.e. the callouts for figures usually should appear sequentially.

Response: Thank you for pointing this out. We have corrected the issue and made sure the figures appear sequentially.

Reviewer #2 (Remarks to the Author): with expertise in cancer immunology, tumor vasculature

The manuscript by Zhu et al. addresses an important question regarding the role of endothelial cell specialization in anti-tumor immune responses. The authors propose that the immunosuppressive tumor microenvironment results from a paucity of tumoral venous endothelial cells. They further show that endothelial cell specific overexpression of COUP TFII is sufficient to promote venous endothelial cell expansion, increase infiltration of T lymphocytes, and decrease growth of tumors in two different models. Overall, the work is of interest and importance for the field.

Response: We appreciate the reviewer's comments on the interest and importance of this work.

I have the following comments:

1. The COUP TFII overexpression model needs to be characterized in more details.

- What are the levels of COUP TFII overexpression in different tumor endothelial compartments, i.e. capillaries, venous endothelial cells and arterial endothelial cells?

- Is such expression heterogeneous or uniform?

- Are newly generated venules located intratumorally or peritumorally?

- What is the impact of COUP-TFII overexpression on vascular density, pericyte recruitment and tumor hypoxia?

Response: First, we confirmed the efficiency of COUP-TFII overexpression in all endothelial cells. Unfortunately, due to technical issues, P-selectin staining did not sustain the permeabilization that is required for nuclear COUP-TFII staining. However, when we compare ICAM1^{high} vs. ICAM1^{low} BECs (a marker associated with venular identity and function), ICAM1^{high} cells do express higher levels of COUP-TFII expression, suggesting that higher level of ectopic COUP-TFII correlates with venular properties. Part of these data are included in **Figures 2A-B**.

Reviewer 2 Point 1a

Second, COUP-TFII-expressing ECs can be observed intratumorally and peritumorally. In addition, we used MAdCAM1 as a marker for newly generated venular ECs (BECs in iCoup but not in control tumors express MAdCAM). We observed MAdCAM1+ ECs both in the keratin+ area and outside the keratin+ area,

suggesting that the venules are both in the tumor nets and in the stroma. These data are included in **Figures S3A-B**.

Reviewer 3 Point 2

Third, iCoup did not significantly change the area of tumor tissue covered by CD31+ vessels. We did not observe a significant change the coverage of vessels by pericytes. Tumors did become more hypoxic, consistent with the compromised function of capillary ECs. These data are included in **Figures S5A-5E**.

Reviewer 2 Point 1c

2. The impact on capillary endothelial cells is well described, what was the effect of COUP TFII overexpression on arterial endothelial cells?

Response: We do not believe that COUP-TFII substantially alters the fate of macro-vessels, including the arteries and large veins. We previously performed scRNAseq profiling in mouse lymphoid tissues in iCoup mice (Dinh *et al.*, 2022). These data suggest that arterial lineages are preserved despite the ectopic COUP-TFII expression.

Arterial endothelial cells are rare and difficult to detect in the tumor tissue. Our attempt to stain for tumor arterial ECs using Connexin 40 was not successful. However, in normal tissues adjacent to the tumor, we were able to discern arterial ECs by imaging, based on CD31 and smooth muscle actin co-staining and the morphological features. Similar to the vasculature in the control mice, iCoup mice also demonstrated abundant arterial structures, despite co-expression of ectopic COUP-TFII. The images are now included in **Figure S3C**.

3. Do the authors observe emergence of HEVs in their models?

Response: We added new data suggesting that ectopic COUP-TFII expression did not transform tumor BECs into fully mature PNA^d-expressing HEVs in our tumor models: we did not detect MECA79 staining in the tumors of iCoup mice. However, we did see upregulated expression of α2-3-sialylated, α1-3-fucosylated Sialyl Lewis X (as stained for by the F2 antibody), suggesting that COUP-TFII induction could promote the synthesis of L-selectin binding glycotypes. The functional consequence of these epitopes remains to be dissected. These have been added to **Figures 20-Q** and in the Discussion.

Reviewer 2 Point 3

4. In many experiments the authors analyse three tumors per condition. Although the results are statistically significant, such low number makes difficult to judge the robustness of the observed effect.

Response: Thank you for the important reminder. We have substantially increased the sample sizes throughout the manuscript. Robustness of reproducibility is also addressed.

Minor comment

1. error bars should be shown as standard deviations, not SEM.

Response: We clarified in figure legends that the error bars are SEM. The variation in the data is shown by including the individual data points. As our goal is to estimate the means and compare them between conditions, we show the standard errors of the means.

Reviewer #3 (Remarks to the Author): with expertise in tumor vasculature

The manuscript entitled 'Tumor immune evasion abrogated by COUP-TFII reprogramming of the vasculature' by Zhu et al, presents evidence that molecular reprogramming of endothelium by COUP-TFII expression supports anti-tumor T cell responses. The authors demonstrate that endothelial expression of COUP-TFII, a transcription factor that drives venous development, induces postcapillary venules in tumors, which facilitates recruitment of T cells into the tumor and suppresses tumor growth, and improves responsiveness to immunotherapy. Current immunotherapy is successful only in a subset of cancer patients, and strategies to enhance anti-tumor immunity in immunologically "cold tumors" are warranted. Here the authors provide an important contribution to this area.

The manuscript presents strong in vivo data that promotion of post-capillary venule phenotype in ECs rendered by COUP-TFII facilitates T cell migration into tumors. These findings are replicated in several models, lending credibility. Nevertheless, some conclusions deserve better/additional supporting evidence to make author's reasoning stronger. The specific points are summarized below.

Response: We appreciate the many positive comments from this reviewer, including the importance of this work and the strength and credibility of our data.

Major comments

1. The authors present in vivo evidence that mice expressing COUP-TFII in endothelial cells (iCoup mice) facilitate enhanced T cell uptake into tumors. However, given the complexity of the tumor microenvironment (TME), these proofs are indirect. An assay directly demonstrating the functional role of endothelial cells in this process, e.g. in isolated endothelial cells in vitro, is required.

Response: We have addressed this point directly in several ways. First, to definitively demonstrate the role of blood endothelial cells in iCoup mice, we selectively blocked selectins (P- and E-selectin) specifically expressed in BECs but not in other cells. By scRNAseq we confirmed the selective expression of P- and E-selectin (*Selp* and *Sele*) in blood endothelial cells, but not in other cells, in the tumor models used in our manuscript. Second, we overexpressed COUP-TFII (*Nr2f2*) in bEND3 cells, a capillary endothelial cell line routinely used for in vitro studies, and assessed the impact on endothelial regulation of T cell migration. In this setting, the overexpression of COUP-TFII correlates with significantly increased T cell transwell migration, consistent across 5 independent repeats, lending support to the in vivo findings that iCoup-driven EC remodeling enhances T cell infiltration.

2. Increased ICAM1 and VCAM1 expression in COUP-TFII expressing tumor endothelial cells is shown by flow cytometry measurement. While quantitative, this approach lacks spatial resolution. Is the vascular expression of ICAM1 and VCAM1 uniform in the TME or are there specific areas with increased local expression of these proteins in WT vs iCoup mice?

COUP-TFII-expressing ECs can be observed intratumorally and peritumorally. In addition, we used MAdCAM1 as a marker for newly generated venular ECs (as shown in Figure 2, BECs in iCoup but not in control tumors express MAdCAM). We observed MAdCAM1+ ECs both in the keratin+ area and outside the keratin+ area, suggesting that the venules are both in the tumor nests and in the stroma. These data are included in **Figures S3A-B**.

(We attempted ICAM1 and VCAM1 staining, which is abundantly expressed outside the EC compartment, making the venular features difficult to visualize, while MAdCAM1 is the best marker for *de novo* post-capillary venular ECs that we have tried.)

Reviewer 3 Point 2

3. The results of a homing assays depicted in Figure 5A, 5B, and 5C, are based on very few cells. It is important that these experiments are reproduced across at least three independent cohorts of mice to ensure reliability of the results.

Response: Thank you for pointing these out. Due to the technical limitations of homing studies and the “cold” nature of the tumor models, the frequencies of homed lymphocytes are indeed limited. We made sure these data were robustly reproducible and updated the figures with substantially increased sample size. In addition, we used 3 different experimental systems of *in vivo* homing, all of which showed increased recruitment of T cells to the tumor tissues in iCoup mice, further strengthening the conclusion.

4. In Figure 5E, endothelial remodeling is shown to influence the population of B cells in the blood. However, data on the B cell population within tumors is not shown. Are B cells changed also in the tumor? Is this functionally important?

Response: Despite the increased abundance of B cells in circulation, the frequency of tumor-infiltrating B cells was not significantly altered by iCoup. These findings contrast to the increased tumor-infiltrating T cells despite the lack of T cell increases in circulation, further suggesting that the increased T cells in iCoup is not due to an overall source of these cells, but rather a result of an actively orchestrated recruitment efforts.

5. Figure 1L shows a change in the ratio of T cell infiltration per P-selectin positive endothelial cells between small and large tumors, suggesting a potential P-selectin independent effect on T cell infiltration. However, an unchanged ratio would be anticipated if P-selectin positive endothelial cells were solely responsible for T cell infiltration. Can the authors provide a better explanation of this figure? Accordingly, please correct the conclusions drawn from this data in the text (lines 90-91).

Response: This is a very good point. We used P-selectin because its bimodal staining patterns allowed us to distinguish capillary EC from ECs enriched with venular properties. However, many other molecules play important roles in the regulation of leukocyte trafficking, including ICAM1 (which also decreased in larger tumors), VCAM1, E-selectin, PECAM, VE-Cadherin, and a myriad of cytokines and chemokines. The decreased T cell-to-P-selectin+ EC ratio reflects the compromised in venular functions that went beyond P-selectin expression.

Reviewer 3 Point 4

6. For all figures, the number of independent experiments and the number of mice used in each experiment must be indicated in the legend.

Response: We made sure to indicate the number of independent repeats and the number of mice used in each experiment.

7. The “Materials and Methods” section is incomplete. Please add information for the following experiments:

- a. Immunofluorescence (Figure 4L) and FACS analysis of blood samples (Figure 5D and E).
- b. Detailed information regarding the data analysis of single-cell RNA sequencing, including specification of the algorithms and settings used. Also, the authors must upload their sequencing data of the scRNAseq to a dedicated repository (ArrayExpress), and provide an accession number in the methods section.
- c. Detailed information for the mouse models, such as genetic background, sex and age distribution.
- d. Method section for statistical analysis is missing.

Response: We have provided more detailed information in the Materials and Methods sections.

8. The “Discussion” section should be extended with a more thorough discussion of results in the context of existing literature, such as by Gabrielle Bergers and Jean-Philippe Girard.

Response: We have added a more thorough discussion and referenced work by Drs. Bergers and Girard.

Minor comments

1. In Figure 2G and text line 121, the name of the capillary marker Podocalyxin is misspelled. Please correct.

Response: Thank you for pointing out the error. We made sure the spelling of Podocalyxin is correct in the text and figure.

2. Figure 4 M-R is erroneously referenced as Figure 5 (lines 176-177). Additionally, treatment compound used in Figure 4 M-R should be specified.

Response: We used the DC101 antibody. This information is updated in the figure and legend. We also corrected the references.

3. In Figure 4L, scale bars are missing.

Response: Scale bars are now included in all the imaging figures.

4. In all the figures containing gating strategies (5D, S2A-C), authors should ensure that the text describing gate populations does not overlap with other texts in the figure.

Response: The gating and labeling are updated accordingly.

5. The legend for Figure 5C is incomplete. Please correct.

Response: The legend is complete.

6. Still in Figure 5C, the authors should indicate in the graph that the population of CD8+ T cells is shown as was done in Figure 5A and 5B.

Response: The figure has been updated accordingly.

7. In Figure 6A please use cubic millimeters (instead of centimeters) to maintain consistency among the figures showing tumor volume.

Response: We made sure cubic millimeters were used consistently to denote tumor volume throughout the manuscript.

8. In Figure 6E, statistics are missing. Please adapt the related text and conclusions if there was no significant difference.

Response: The legend is complete with information on statistical significance.

9. A control group treated with depleting antibodies should be included in Figure 6I. Also, please explain why different depleting strategies were used in Figure 5F and 5I.

Response: In our new Figure we updated the data with CD8 depletion to maintain consistency with Figure 5.

10. Authors should use the same group names across related figures, e.g. Figure 7A -7C and 7D – 7F. Also, names/labels of graph axes and conditions should be used consistently throughout the manuscript.

Response: We corrected the labeling of axes and conditions to maintain consistency throughout the manuscript.

11. In Figure 7C and 7F it is unclear which treatment groups are being compared for statistical evaluation. Please re-align the vertical lines.

Response: We re-aligned the vertical lines.

12. Can the authors explain why both CD4+ (Figure 7A) and CD8+ (Figure 7B) T cells are shown in some experiments, while only the entire population of T cells is shown in others (Figure 7D)? If possible, please show the same populations to facilitate data interpretation.

Response: In the new Figure we show both CD4+ and CD8+ T cells in all experiments.

13. The text (lines 83-84) and figure legend associated with Supplementary Figure S1A do not match the displayed graphs. Please correct.

Response: We corrected the information.

14. In Supplementary Figure S1 E and F please add the correlation coefficients.

Response: We added the correlation coefficients.

To all reviewers:

Thank you for your time and efforts in providing a second round of review to improve our paper. Here we provide a revised manuscript addressing your concerns. **Black** is the reviewer's comments and **Blue** is our response. Changes in the manuscripts are highlighted with a bar on the left side of the text.

Reviewer #1 (Remarks to the Author): with expertise in cancer immunology, tumor vasculature

A very nice study. This reviewer's comments have been adequately addressed.

Minor comments:

On line 32 do you mean "tumor nests" rather than "nets"?

In fig S3, "tumor nests" or "tumor nets"?

Response: Thank you for pointing out this mistake. We changed to the correct term "tumor nests".

On line 246 should be "...recruited into the tumor tissue."

Response: Thank you. We corrected the grammatical error.

On line 290 is "CD8beta-depleting" correct?

Response: Yes, we used depleting CD8 beta-depleting antibodies to specifically deplete CD8+ T cells, which express CD8 $\alpha\beta$, while preserving CD8+ dendritic cells, which only express the CD8 α chains.

Reviewer #3 (Remarks to the Author): with expertise in tumor vasculature

The authors addressed most of the comments. However, a few points remain that might have been missed and need to be addressed.

Most importantly, while the Material and Methods section has been updated, detailed information on the analysis of scRNA-seq data, including the specific algorithms and settings used in the analysis, is missing. Without this information the analysis cannot be reproduced. For the same reason, as requested previously, the authors need to upload the scRNA-seq data to a suitable repository (such as GEO, ArrayExpress) and include the accession number in the methods section.

Response: We have deposited the raw scRNAseq data in GEO, accession number [GSE281284](https://www.ncbi.nlm.nih.gov/geo/query/acc.cgi?acc=GSE281284), which is included in the newly added "Data Availability" section in the Materials and Methods. Please access the data with Reviewer token unwtwyiwncjxjap, and the data will be public upon the publication of this paper.

In addition, we revised the Materials and Methods to include algorithms and packages used for the analyses, as well as the parameters we used.

Minor unsolved issues are:

1. The authors indicated that the legend for graph S1A had been revised, but it seems to be unchanged. Please update the legend to ensure that it clearly describes all three figures.

Response: We have improved the figure legends to accurately reflect the three figure panels in Figure S1A. The legend now reads "Flow cytometric quantification of the abundance of BECs, the representation of P-selectin+ venules among BECs, and the BEC expression fluorescence intensity of P-selectin in orthotopic KPC tumors."

2. In the legend for Figure 5C, the authors state that "Quantification indicates the frequency of homed donor CD8+ T cells as % of total cells dissociated from the tumor". However, the figure includes four graphs showing the quantification of different cell types.

Response: We edited the figure legend to accurately reflect the data. The legend now reads "Quantification in (B-C) indicates the frequency of homed donor-derived leukocytes as % of total cells dissociated from the tumor."

3. The authors were asked to use consistent group names across related figures, such as Figures 7A-7C and 7D-7F, but Figure 7F was missed in this process.

Response: We fixed the groups names for Figure 7F.

4. For Figure 8, please indicate the number of independent experiments and the number of mice used in each experiment.

Response: We included the number of mice per group and the number of independent repeats in the legend for Figure 8.